# Genomic evidence reveals three W-autosome fusions in *Heliconius* butterflies

**Nicol Rueda-M**[1,2], **Carolina Pardo-Diaz**[1], **Gabriela Montejo-Kovacevich**[3], **W. Owen McMillan**[4], **Krzysztof M. Kozak**[4,5], **Carlos F. Arias**[4,6], **Jonathan Ready**[7,8], **Shane McCarthy**[2], **Richard Durbin**[2,9], **Chris D. Jiggins**[3], **Joana I. Meier**[3,2©], **Camilo Salazar**[1©]*

**1** Biology Program, Faculty of Natural Sciences, Universidad del Rosario, Bogotá, Colombia, **2** Tree of Life Programme, Wellcome Sanger Institute, Hinxton, United Kingdom, **3** Department of Zoology, University of Cambridge, Cambridge, United Kingdom, **4** Smithsonian Tropical Research Institute, Panama City, Panama, **5** Museum of Vertebrate Zoology, Berkeley, California, United States of America, **6** Data Science Lab, Office of the Chief Information Officer, Smithsonian Institution, Washington, Washington DC, United States of America, **7** Institute for Biological Sciences, Federal University of Pará - UFPA, Belém, Brazil, **8** Centre for Advanced Studies of Biodiversity - CEABIO, Belém, Brazil, **9** Department of Genetics, University of Cambridge, Cambridge, United Kingdom

© These authors contributed equally to this work.

* camilo.salazar@urosario.edu.co

**Data Availability Statement:** The sequence data generated in this study has been deposited in the European Nucleotide Archive (ENA) under the accession code PRJEB60296, or elsewhere in the

## Abstract

Sex chromosomes are evolutionarily labile in many animals and sometimes fuse with autosomes, creating so-called neo-sex chromosomes. Fusions between sex chromosomes and autosomes have been proposed to reduce sexual conflict and to promote adaptation and reproductive isolation among species. Recently, advances in genomics have fuelled the discovery of such fusions across the tree of life. Here, we discovered multiple fusions leading to neo-sex chromosomes in the *sapho* subclade of the classical adaptive radiation of *Heliconius* butterflies. *Heliconius* butterflies generally have 21 chromosomes with very high synteny. However, the five *Heliconius* species in the *sapho* subclade show large variation in chromosome number ranging from 21 to 60. We find that the W chromosome is fused with chromosome 4 in all of them. Two sister species pairs show subsequent fusions between the W and chromosomes 9 or 14, respectively. These fusions between autosomes and sex chromosomes make *Heliconius* butterflies an ideal system for studying the role of neo-sex chromosomes in adaptive radiations and the degeneration of sex chromosomes over time. Our findings emphasize the capability of short-read resequencing to detect genomic signatures of fusion events between sex chromosomes and autosomes even when sex chromosomes are not explicitly assembled.

## Author summary

Fusions between sex chromosomes and autosomes are thought to have the potential to resolve sexual conflict and enhance local adaptation or reproductive isolation between species. Here, we discovered such fusions in *Heliconius* butterflies. These butterflies mostly have a very stable karyotype with 21 chromosomes and high synteny across

ENA if obtained from previous studies, as specified in S1 Table for each individual. Reference genomes for the species *H. sara, H. congener*, and *H. sapho* can be found at https://tolqc.cog.sanger.ac.uk/. All data associated with the individuals used in this study are detailed in S1 Table.

**Funding:** This work was supported by the Universidad del Rosario (https://urosario.edu.co/) (BigGrant IV-FGD005 and Fondos Concursables IV-FPD004 to CS and CP); Colombian Ministry of science, technology and innovation (https://minciencias.gov.co/) (MinCiencias – doctoral scholarship 727 to NRM); Branco Weiss – Society in Science fellowship, Royal Society University Research Fellowship and Wellcome Trust award (https://brancoweissfellowship.org/, https://royalsociety.org/ and https://wellcome.org/) (URF \R1\221041 and 220540/Z/20/A to JIM). The funders had no role in study design, data collection and analysis, decision to publish, or preparation of the manuscript.

**Competing interests:** The authors have declared that no competing interests exist.

species. However, the five species in the *sapho* subclade have up to 60 chromosomes, suggesting that they have undergone many chromosomal fissions. We document that in addition to the fissions, the *sapho* subclade also shows multiple fusions between the female-specific sex chromosome W and autosomes. We found an ancestral W-4 fusion shared by all five species and additional W fusions with chromosomes 9 and 14 shared by two species each. Even though in many *sapho* subclade species the autosomes have undergone fissions, the chromosomes fused with the W did not undergo fissions. Our study reveals the power of short-read sequencing to detect the genomic signatures of Sex-A fusions and shows *Heliconius* butterflies as a promising system for studying the causes and consequences of sex chromosome evolution.

## Introduction

Sex chromosome-autosome (Sex-A) fusions contribute to the evolution of neo-sex chromosomes [1,2], but it remains unclear what promotes them. Sexually antagonistic selection, direct selection, genetic drift, meiotic drive, and sheltering of deleterious mutations have all been suggested as possible drivers of Sex-A fusions [3–6]. Sexually antagonistic selection is thought to favour the fusion of sex chromosomes with autosomes harbouring genes under sexually antagonistic selection [7]. There is limited evidence for this hypothesis e.g. in sticklebacks [8], *Drosophila* flies [9], warblers [10] and butterflies [11]. Sex-A fusions can also become fixed due to meiotic drive (including holocentric drive in holocentric organisms) [12], such as female meiotic drive elements on W/Z-A fusions that preferentially end up in the egg instead of the polar bodies [13]. An alternative hypothesis is deleterious mutation sheltering, when Sex-A fusions are favoured because they prevent the expression of recessive deleterious alleles in the heterogametic sex [3]. As with other chromosomal rearrangements, Sex-A fusions can reduce recombination and potentially strengthen reproductive isolation [14,15]. For instance, in the Japanese threespine stickleback *Gasterosteus aculeatus*, a Sex-A fusion resulted in a neo-X chromosome that linked loci underlying behavioural isolation traits and hybrid sterility [4]. Sex-A fusions may also facilitate adaptation, such as the Sex-A fusion in *Cydia pomonella* (Tortricidae), which apparently linked two insecticide-resistance genes and genes involved in detoxifying plant metabolites [16].

Cytogenetic and genomic studies revealed that Sex-A fusions have occurred many times across vertebrates [6,17,18], and invertebrates such as spiders [19,20], *Drosophila* flies [21,22] or true bugs of the genus *Dysdercus* [23]. In Lepidoptera (butterflies and moths), examples of Sex-A fusions include *Danaus* [24,25] and *Leptidea* butterflies [26,27], among others [16,28–31]. Compared to taxa with a single centromere per chromosome, the holocentric chromosomes of Lepidoptera may facilitate the establishment of fusions as they are less likely to cause segregation problems during cytokinesis and thus reduce hybrid fitness [32]. Nonetheless, butterflies and moths have remarkably constrained chromosome evolution [33,34] with most species having a ZW or Z0 sex determination system and a haploid chromosome number ranging between 28 and 32, except for few groups that have experienced extensive fission and fusion events [30,35].

Here, we focused on *Heliconius* butterflies, which have undergone 10 ancestral fusions and thus display 20 autosomes, along with Z and W sex chromosomes with high collinearity across species [36]. Only a few species in the genus differ in this ancestral chromosome number, especially species in the *sara/sapho* clade, with some having up to 60 chromosomes [37]. The *sara/sapho* clade comprises 12 species [38] that are different from other *Heliconius* due to their

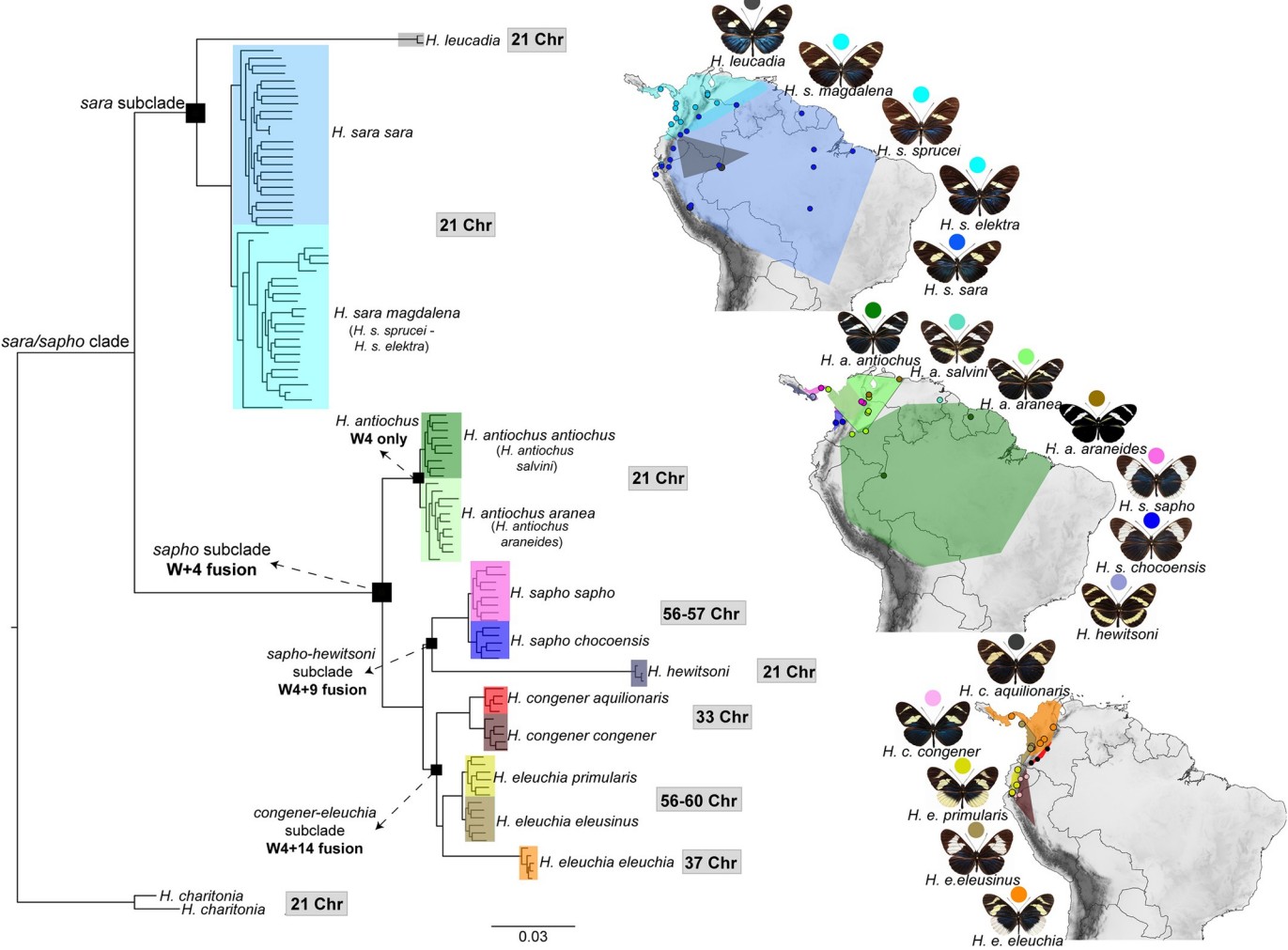

**Fig 1. Phylogeny and distribution of the *sara/sapho* clade.** Genome-wide ML phylogeny, highlighting the position of the *sara* and the *sapho* subclades. Bootstrap support values were 100 for all branches at species level. The distribution of all 17 subspecies in the clade from [44,45] is shown, where dots represent sampling locations in the present study. Each of the 17 subspecies is represented by a single colour, that is the same in the phylogeny and the distribution. The chromosome number of each species is shown in grey rectangles [37]. The species' distribution map was created using the free and open source QGIS (44), with a base shapefile obtained from https://www.naturalearthdata.com/downloads/10m-cultural-vectors/ and an elevation raster from https://csidotinfo.wordpress.com/data/srtm-90m-digital-elevation-database-v4-1/.

inability to synthesize cyanogens (compounds that deter predators) thus forcing them to rely on sequestered plant toxins [39]. A subclade of five species (*sapho* subclade; Fig 1) within the *sara/sapho* clade shows particularly high diversification rates [38] and a high number of chromosomes [37].

We assembled a reference genome of *Heliconius sara magdalena*, a species with typical karyotype of 21 chromosomes. We found that it was fully collinear with the chromosome-level assemblies of *H. melpomene*, *H. erato* and *H. charithonia*. Next, we generated whole-genome resequencing data from 114 individuals of all seven species of the *sara/sapho* clade with high diversification rates to completely resolve their phylogenomic relationships and study genomic differences between the species. We discovered that the *sapho* subclade, which shows a larger number of chromosomes in some species, exhibits fusions between the W chromosome and

autosomes (W-A fusions). One W-A fusion is shared by all five species, whereas two additional W-A fusions are shared by two species each. Interestingly, while in some species the autosomes have been split up into many chromosomes, the autosomes fused to the W have not undergone fissions. These W-A fusions make the *Heliconius sapho* subclade a prime study system for the evolution of neo-sex chromosomes.

## Results

### High-quality reference genome assembly for *Heliconius sara magdalena*

We assembled the genome of *Heliconius sara magdalena* using two laboratory-reared females from a stock population originating from Panama (S1 Table). Using the first individual (Bio-Sample SAMEA8947140), we obtained 24 Gbp PacBio CLR data with a 50x coverage and an N50 subread length of 16.5 kb and 122 Gbp of 10X linked-read Illumina data with a 294x coverage. From the second individual (BioSample SAMEA8947139), we obtained 137 Gbp Hi-C Illumina data with a 111x coverage. The final assembled genome consists of 348.8 Mbp in 384 scaffolds. The contig N50 was 8.2 Mbp and that of scaffold N50 was 17.8 Mbp. Our genome showed the highest contiguity (S2 Table) and BUSCO statistics (S3 Table) of *Heliconius charithonia*, *Heliconius erato* and *H. melpomene* genomes. The BUSCO completeness, using the Lepidoptera gene set, achieved 98.2% single-copy BUSCOs and fewer duplicated, fragmented, and missing BUSCOs than in the genomes of *H. erato* [40], *H. melpomene* [41] and *H. charithonia* [42] (S3 Table). We assigned the largest 22 scaffolds to 20 autosomes (one scaffold for each chromosome, except for chromosome 11 which is composed of three scaffolds) and one scaffold to the Z chromosome based on synteny with the *Heliconius melpomene* genome. *H. sara* chromosomes are collinear with this genome, as well as with *H. erato* and *H. charithonia* (S1 Fig). For more information on the genome see https://tolqc.cog.sanger.ac.uk/durbin/jiggins/Heliconius_sara/ and https://www.ncbi.nlm.nih.gov/datasets/genome/GCA_917862395.2/.

The W chromosome did not assemble well, as is commonly seen in lepidopteran genomes [30]. We thus used whole genome resequencing data from 114 individuals collected in this study (see below "Whole-genome resequencing dataset section") to assign scaffolds to the W chromosome based on sequencing depth differences between males and females. Among the 360 scaffolds not assigned to a specific chromosome, 32 exhibited a higher mean depth in females than in males in *Heliconius sara*. This pattern suggests that these 32 scaffolds likely constitute a part of the W chromosome (S2 Fig). Interestingly, the reads from the *sapho* subclade species (*H. antiochus*, *H. sapho*, *H. hewitsoni*, *H. eleuchia*, and *H. congener*) did not align to these 32 scaffolds, suggesting that W chromosome of these species is either too divergent from the *H. sara* W chromosome (S2 Fig) or not present. These putative W scaffolds in *H. sara* correspond to a single homolog in *H. charithonia* (S1C Fig).

### Whole-genome resequencing dataset

A total of 114 individuals were successfully whole-genome resequenced. Our dataset exhibits high taxonomic completeness covering all 7 species within the *sara/sapho* subclade (*H. sara*, *H. leucadia*, *H. antiochus*, *H. sapho*, *H. hewitsoni*, *H. eleuchia*, and *H. congener*) and 19 out of the 28 described subspecies [43] (S1 Table). The average mapping percentage to the *H. sara* genome was 95.56% (range: 77.38%–99.14%) (S1 Table). We observed a strong phylogenetic signal in the mapping proportion and, consequently, in the proportion of missing data per individual (S1 Table and S3 Fig). The mapping proportion was 97.7%, 97.6%, and 96.4% for *H. sara* specimens, its sister species *H. leucadia*, and the *sapho* subclade, respectively. One *H. congener* and one *H. antiochus* individual exhibited a particularly high proportion of missing data

(19.5% and 39.5%, respectively) and low mean depth of coverage (7.4X and 9.1X, respectively) (S1 Table and S3 Fig), and were thus excluded from further analyses.

## Phylogenetic analysis reveals two main subclades and uncovered incongruence across the genome

We reconstructed a Maximum Likelihood (ML) phylogenetic tree using all 112 individuals and 183,282,470 concatenated sites. This phylogeny separated individuals into two main subclades, consistent with the PCA analyses (S4 Fig): (i) *sara* and (ii) *sapho* (Fig 1). The *sara* subclade is composed of two species, namely *H. sara* and *H. leucadia*, where *H. sara* is subdivided into an Andean group (*H. s. magdalena*, *H. s. sprucei*, and *H. s. elektra*) and an Amazonian group (*H. s. sara*). The *sapho* subclade was split into two well-resolved lineages (*H. antiochus* and a subclade composed of two monophyletic groups: *H. eleuchia/H. congener* and *H. sapho/ H. hewitsoni*). *H. antiochus* appeared as a monophyletic group split into an Andean group (*H. a. aranea* and *H. a. araneides*), and an Amazonian group (*H. a. antiochus* and *H. a. salvini*) (Fig 1). *H. antiochus* nested into the *sapho* subclade, whereas *H. hewitsoni* was found to be sister to *H. sapho*, thus resolving the previously undetermined position of these species [38].

To complement the concatenated phylogeny, we also reconstructed a species tree with Astral using males only. The species phylogeny closely mirrored the genome wide phylogeny (S5A Fig). Although *H. hewitsoni* and *H. sapho* were recovered as sister species, their branch lengths (measured in coalescence units) (S5A Fig) were short, indicating high levels of discordance. This was also the case for the subspecies of *H. eleuchia*, where *H. e. eleusinus* and *H. e. primularis* sometimes group with *H. congener* (S5A Fig). These phylogenetic discordances were also evident in the DensiTree analysis visualising 271 phylogenies together (S5A–S5B Fig).

We found strong phylogenetic incongruence across chromosomes. The whole-genome topology was recovered on only eight chromosomes (S6–S27 Figs), while nine chromosomes showed *H. congener* appearing as sister either to *H. e. eleuchia* or to a clade composed of *H. e. eleusinus* + *H. e. primularis* (S6–S27 Figs). Similarly, *H. hewitsoni* was not sister to *H. sapho* on eight chromosomes (S6–S27 Figs). Interestingly, we observed sex-specific clustering on three chromosomes (S27B Fig). On chromosome 4 (Chr4), all species in the *sapho* subclade (*H. antiochus*, *H. sapho*, *H. hewitsoni*, *H. eleuchia*, and *H. congener*) showed females and males forming separate clades within each species, whereas the males of *H. congener* and *H. eleuchia* formed a shared clade and their females formed a shared clade (S27B and S9 Figs). Sex-specific clades were also observed on chromosome 9 (Chr9) in the *sapho-hewitsoni* subclade (S27B and S14 Figs) and on chromosome 14 (Chr14) in the *congener-eleuchia* subclade (S27B and S19 Figs).

## Haplotype-based phylogenetic analysis on chromosomes 4, 9 and 14

The grouping by sex we observed in the phylogenetic trees of Chr4, Chr9 and Chr14 suggests possible fusions between these autosomes and either the Z or W chromosome, or possibly both (females of *Heliconius* are ZW and males are ZZ) [43]. As females of Lepidoptera lack crossing over and their meiosis is achiasmatic, they do not recombine [46]. This means that if the W chromosome is involved in the fusion (Fig 2A), the Sex-A fusion would be restricted to females and the fused chromosome would tend to accumulate mutations and/or structural variants leading to divergence from its unfused homologue. The unfused Chr4 would become a neo-Z2 chromosome in all species of the *sapho* subclade, Chr9 would become a neo-Z3 in the *sapho-hewitsoni* subclade, and Chr14 would become a neo-Z3 in the *congener-eleuchia* subclade. Alternatively, if the Z chromosome is involved in the fusion (Fig 2B), females would initially still have the unfused homologue (neo-W) that would start to accumulate mutations

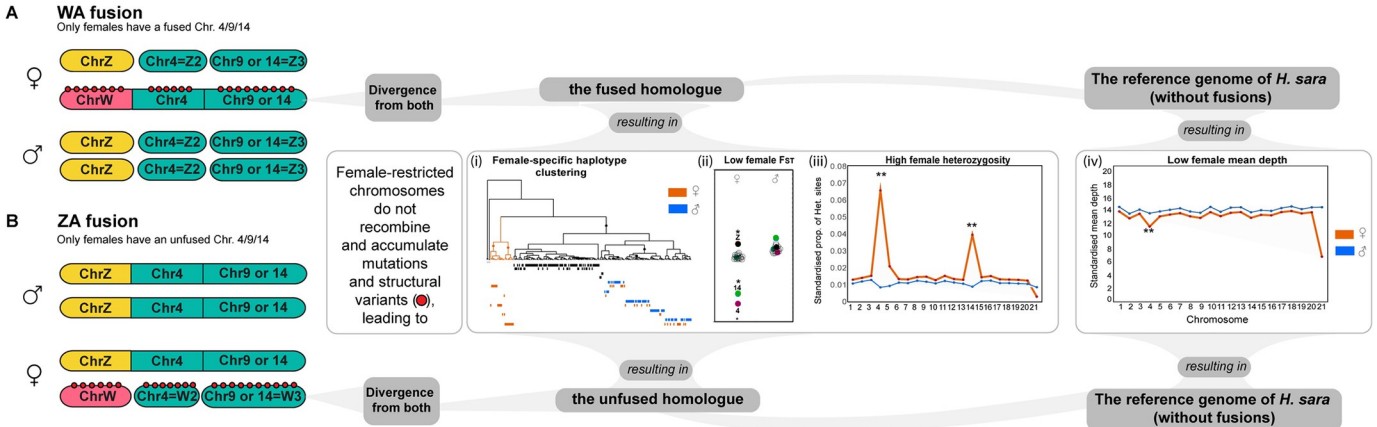

**Fig 2. Scenarios of Sex-A fusions involving either the W or Z chromosomes.** Scenario of (A) WA fusion and (B) ZA fusion for Chr4, Chr9 and Chr14 differentiating the expected pattern by sex: (i) Female-specific haplotype clustering, (ii) low female $F_{ST}$, (iii) high female heterozygosity and (iv) low female mean depth. In both fusion scenarios, we would also expect that the unfused chromosomes 4/9/14 would become neo-Zs in the case of W-A fusions or neo-Ws in the case of Z-A fusions. For more details see main text. Yellow: Z chromosome. Pink: W chromosome. Green: autosome. Dark-grey semicircles: fusions. Red dots: mutations and/or structural variants. Lastly, it is possible that both the W and Z chromosomes may have fused with these autosomes, resulting in a neo-W and neo-Z sex chromosome constitution.

and/or structural variants, leading to the divergence from the Z-fused homologue. If the sex-autosome fusion was with the Z in the *sapho* subclade, Chr4 would become neo-W2 in all five species, Chr9 would become neo-W3 in the *sapho-hewitsoni* subclade and Chr14 would become neo-W3 in the *congener-eleuchia* subclade (Fig 2B). Lastly, it is possible that the autosomes fused both with the Z and the W, or that some autosomes fused with the Z and others with the W, leading to neo-Ws and neo-Zs, and reducing the number of chromosomes.

Under either fusion to the Z or the W, we predict i) genealogies where each female has one haplotype that forms part of the same clade as the male haplotypes, while the other haplotype forms a female-specific clade (hereafter called female-specific haplotype clustering), ii) low genetic differentiation ($F_{ST}$) on Chr4, Chr9, and Chr14, due to higher divergence between males and females within populations and lower variation between populations if they share the same sex-autosome fusions (see "Patterns of genetic differentiation"), iii) high proportion of sites where all females are heterozygous due to the presence of two different haplotypes (see "Sex-specific differences in heterozygosity and mean depth"), and iv) low sequencing depth in females due to poor mapping of the female-specific haplotypes that have accumulated mutations and structural variants (see "Sex-specific differences in heterozygosity and mean depth") (Fig 2).

Consistent with our hypotheses, we identified 218,839 SNPs on Chr4 where all males within the *sapho* subclade were homozygous and most females were heterozygous (up to one female per species was homozygous). To study the phylogenetic relationships among male and female haplotypes at these sites, we phased our dataset and inferred marginal phylogenies from ancestral recombination graphs constructed using Relate [47]. We subsampled the dataset to every 1000th SNP with high female heterozygosity (214 SNPs). At 31% of these sites, we recovered the expected marginal phylogeny with a female-specific haplotype clade (Fig 3A). The sites displaying this pattern were distributed across the entire chromosome and were not concentrated in a specific region (Fig 3B). Another set of SNPs (53%) exhibited marginal phylogenies where female haplotypes from at least two species within the *sapho* subclade clustered as expected for a shared Sex-A fusion (S28 Fig). The remaining sites (16%) exhibited a mixed signal similar to

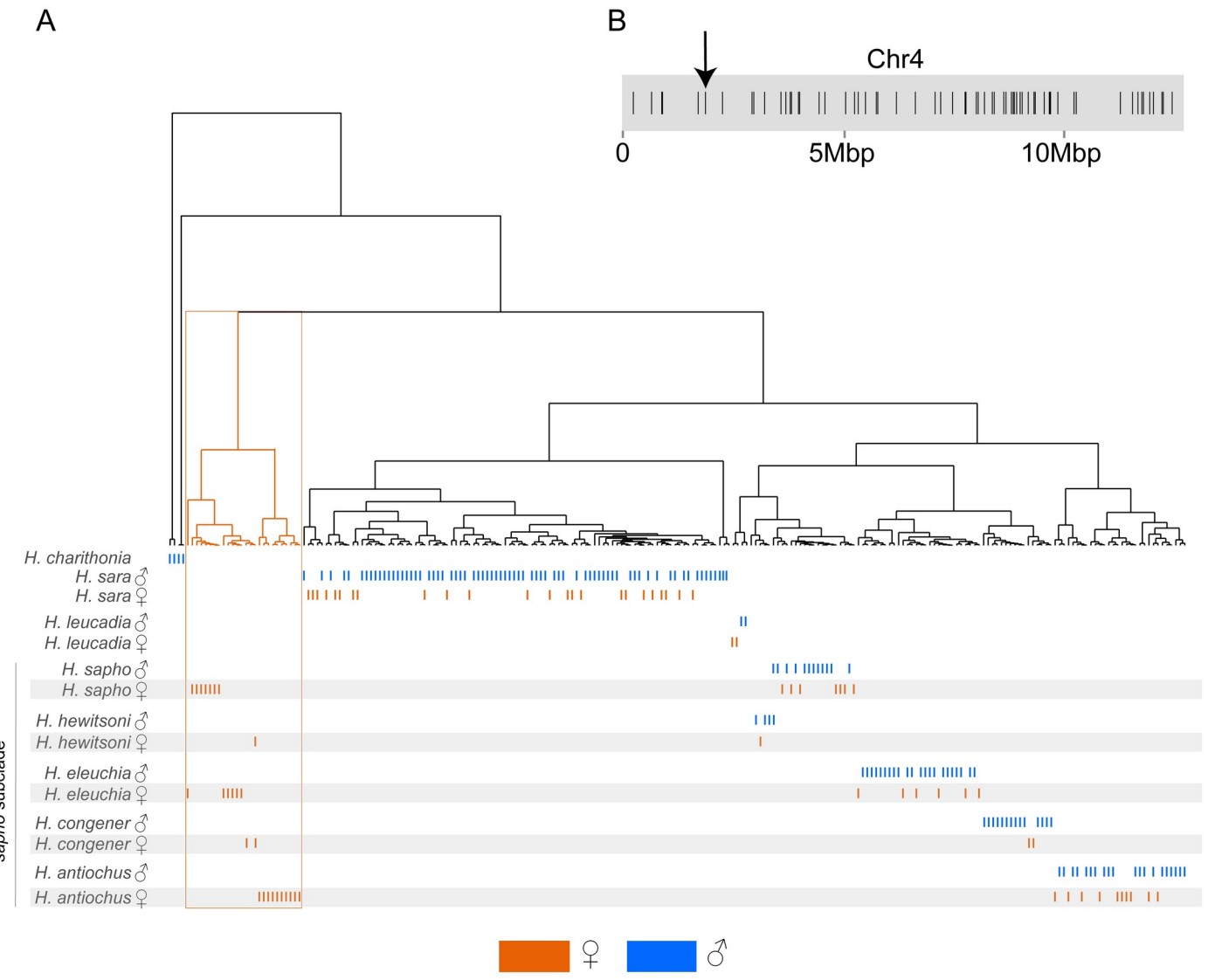

**Fig 3. Marginal tree at a focal SNP inferred from an ancestral recombination graph of Chr4.** Genealogy of the region around SNP 1,900,983 on Chr4 showing a pattern fully consistent with a shared Sex-A fusion ancestral to all five species of the *sapho* subclade; a total of 66 subsampled SNPs across this chromosome showed the same pattern. Each vertical line represents an individual haplotype, and the haplotypes of all individuals are shown differentiating those of females (orange) from those of males (blue). Note that one allele of the females in the *sapho* subclade clustered with the alleles of males, while the other female allele formed a separate group (highlighted in orange). (B) Position of each of the 66 focal SNPs on Chr4 that showed genealogies consistent with a Sex-A fusion; they were not clustered in a specific region but rather distributed along the entire chromosome. The position of the SNP 1,900,983 used for the genealogy in A, is indicated by an arrow.

the unphased phylogenetic tree obtained for the entire chromosome (S27B Fig). This is likely due to phasing errors where the fused and unfused homologues of the females are mixed. On Chr9, we found 66,344 sites where the species in the *sapho-hewitsoni* subclade exhibited the expected pattern given a Sex-A fusion. In this case, we found that the 23% of the 66 sampled sites displayed the expected marginal phylogenies (S29A Fig). The remaining 77% of the SNPs showed a mixed signal as in Chr4 (S27B Fig). Finally, on Chr14 of the species in the *congener-eleuchia* subclade we identified 57,953 sites with the described pattern. Among the 57 subsampled sites, 44% showed the expected genealogical clustering (S30A Fig) while the remaining 56% displayed a mixed signal (S27B Fig).

## Patterns of genomic differentiation

Genomic differentiation ($F_{ST}$) was higher between the putative sister species *H. sapho* and *H. hewitsoni* than between the sister species *H. congener* and *H. eleuchia*, and also higher than the comparison between Andean and Amazonian subspecies in *H. antiochus* (average $F_{ST}$ = 0.33, 0.26, and 0.07 respectively) (S31 Fig). *H. sara* vs. *H. leucadia* were the least differentiated pair (average $F_{ST}$ = 0.05) (S31 Fig). We also observed elevated $F_{ST}$ values on the Z chromosome compared to autosomes in all but one comparison (*H. eleuchia* vs. *H. congener*) (S31 Fig). In line with predictions from sex-autosome fusions in the *sapho* subclade, Chr4 showed lower than average $F_{ST}$ values in this subclade, but not in *H. sara* vs. *H. leucadia* (Figs 4A and S31). This pattern is expected if the females have two highly divergent non-recombining haplotypes (one fused and one unfused), leading to high within-population nucleotide diversity (π) (S32B Fig), even if there is some divergence between populations ($D_{XY}$) (S32A Fig). Chr9 showed lower $F_{ST}$ in *sapho-hewitsoni* subclade (Figs 4A and S31), and Chr14 in the *congener-eleuchia* subclade (Figs 4A and S31). The same pattern was observed when we compared $F_{ST}$ between subspecies (S33–S35 Figs). In line with expectations (Fig 2), the observed pattern of lower $F_{ST}$ on Chr4, Chr9, and Chr14 was exclusive to females and absent in males across all comparisons mentioned above within all five species of the *sapho* subclade (Fig 4B). The observed differences in $F_{ST}$ between males and females were statistically significant for these three chromosomes (Wilcoxon test p < 0.01) (S36 Fig). The $F_{ST}$ values of these chromosomes within females were also significantly lower compared to those of the other chromosomes (Wilcoxon test p < 0.01) (S36 Fig).

## Sex-specific differences in heterozygosity and mean depth

Consistent with our hypotheses (Fig 2), Chr4, Chr9, and Chr14 showed striking sex-specific differences in the proportion of heterozygous sites and mean depth only in species of the *sapho* subclade, supporting three fusions of these chromosomes with the Z or W chromosomes or both (Fig 5). The strongest difference in the proportion of heterozygous sites was observed on Chr4 where females of *H. eleuchia*, *H. congener*, *H. sapho*, *H. hewitsoni* and *H. antiochus* showed a higher proportion of heterozygous sites than males and the other autosomal chromosomes in females (Fig 5B). Females of *congener-eleuchia* subclade also showed a high proportion of heterozygous sites on Chr14 and in *sapho-hewitsoni* subclade on Chr9 (Fig 5B). Differences between males and females were significant on these three chromosomes for all species of the *sapho* subclade (Wilcoxon test, p <0.01) (S37 Fig), except for *H. hewitsoni* where differences could not be tested due to low sample size. The proportion of heterozygous sites of females was also significantly higher for Chr4 than for the other chromosomes in *H. eleuchia*, *H. sapho*, *H. antiochus* and *H. congener* (Wilcoxon test, p <0.01) (S37 Fig). The same was true for Chr14 in *congener-eleuchia* subclade, and Chr9 in *H. sapho* (Wilcoxon test, p <0.01) (S37 Fig). A high proportion of heterozygous sites was not observed in the females of the species *H. sara* and *H. leucadia* on any chromosome (Figs 5 and S37). Chromosome Z exhibited lower proportion of heterozygous sites than autosomal chromosomes in females across all species (Figs 5 and S37).

Females of *H. eleuchia*, *H. sapho*, *H. hewitsoni*, and *H. antiochus* also showed a reduced mean depth on Chr4, whereas the mean depth on that chromosome in males was normal (Fig 5B). However, these differences were only significant in *H. sapho* (Wilcoxon test, p ≤ 0.01) (S38D Fig). This pattern was not true for Chr14 in *congener-eleuchia* subclade, nor Chr9 for *sapho-hewitsoni* subclade (Fig 5). The mean depth of Chr4 was also lower than that of all other autosomes in females of *H. sapho* (Wilcoxon test, p ≤ 0.01) (S37D Fig). However, this was not true for Chr14 and Chr9 (S38 Fig).

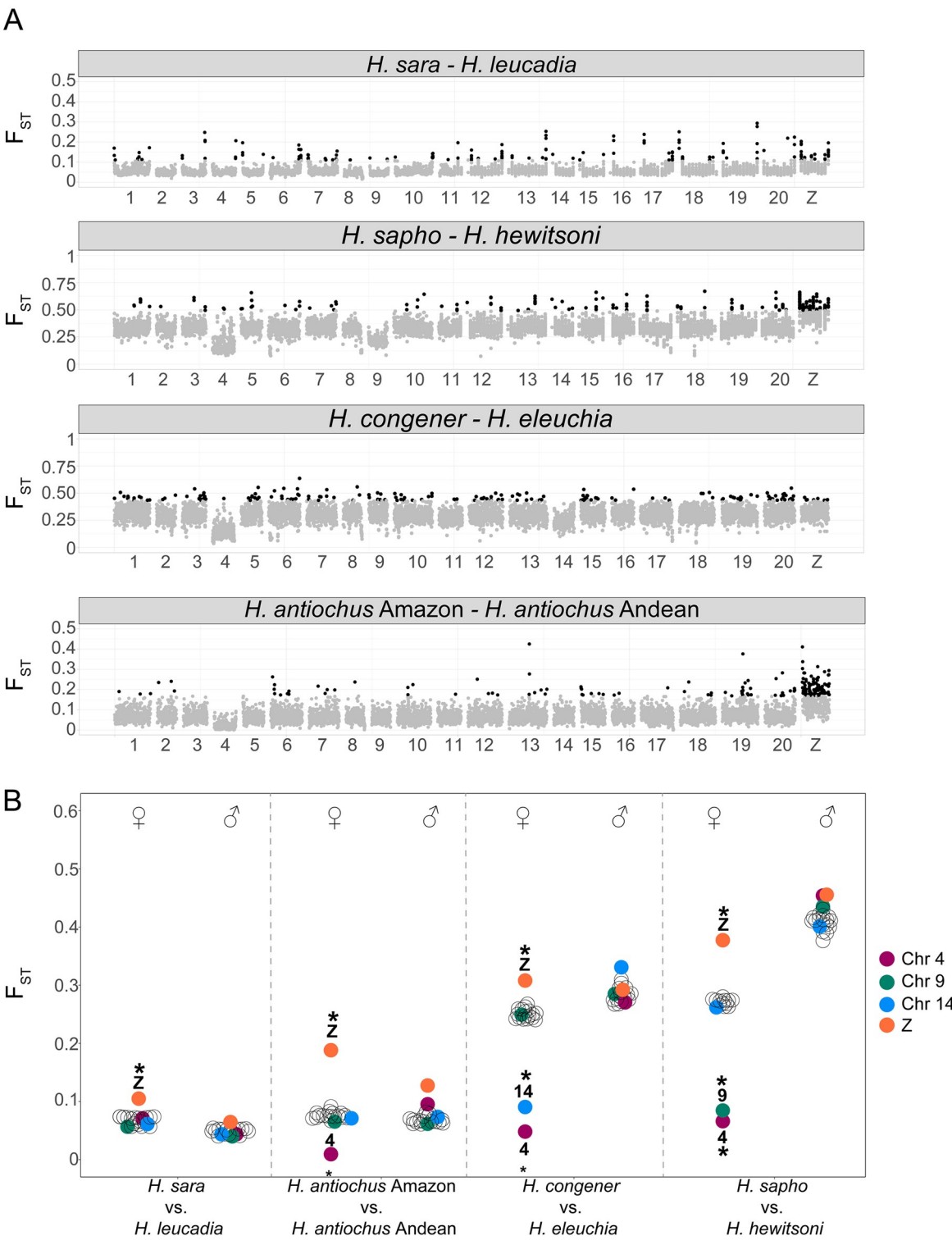

**Fig 4. Genome-wide $F_{ST}$ in the *sara/sapho* clade.** (A) $F_{ST}$ between pairs of species. Each point represents a 50Kb window, whereby the top 5% windows are shown in black. The numbers below correspond to *H. sara* chromosomes. Subspecies comparisons are shown in S32–S34 Figs. (B) $F_{ST}$ between pairs of species by sex. Each circle represents a chromosome, and chromosomes with evidence of Sex-A fusions are colour coded (* indicates outlier chromosomes, $p<0.01$). The observed reduction in $F_{ST}$ in females is due to higher genetic diversity within species due to the divergence between the sex chromosome-fused and unfused haplotypes in females.

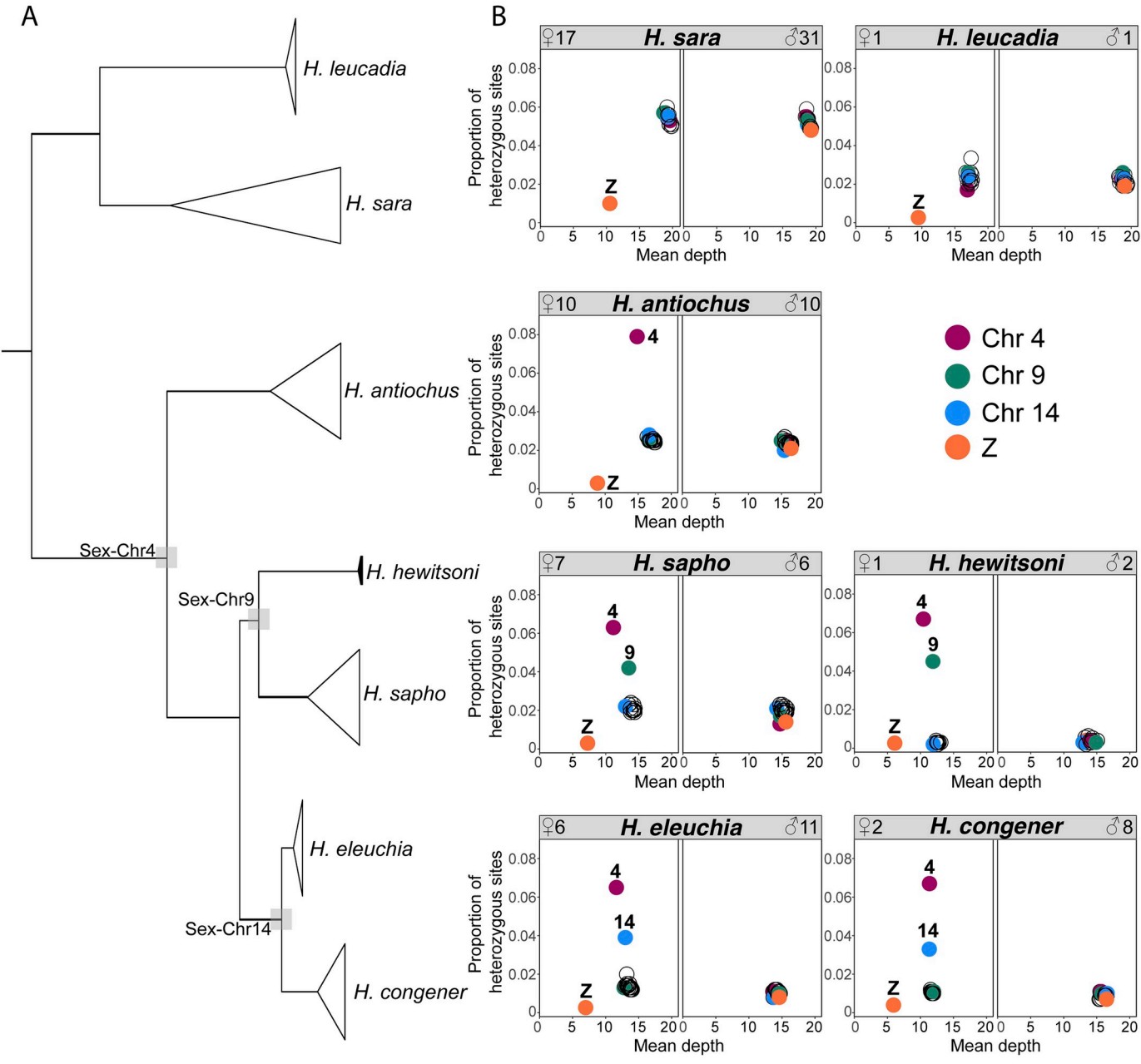

**Fig 5. Genome-wide topology and patterns of heterozygosity and sequencing depth across the genome.** (A) Genome-wide topology with grey squares highlighting nodes with putative Sex-A fusions. (B) The proportion of heterozygous sites vs. mean sequencing depth per chromosome, by sex and in each species. Each circle represents a chromosome, and chromosomes with evidence of Sex-A fusions are colour coded. The high heterozygosity in females is due to the presence of the fusion in only one of the haplotypes, which becomes divergent from its counterpart. The low mean depth is because the haplotype limited to females diverged enough to be difficult to map onto the reference genome of *H. sara*.

The sliding window analyses on Chr4, Chr9, and Chr14 revealed that the excess heterozygosity in females is present in most windows along the entire chromosomes (i.e., it is not concentrated in a specific region on the chromosomes; Wilcoxon test, $p < 0.01$) (Figs 6 and S39–S40). Also, the mean depth values were lower for females than males in most windows on Chr4 for all *sapho* subclade species except *H. hewitsoni*, Chr9 for *H. sapho* and Chr14 for

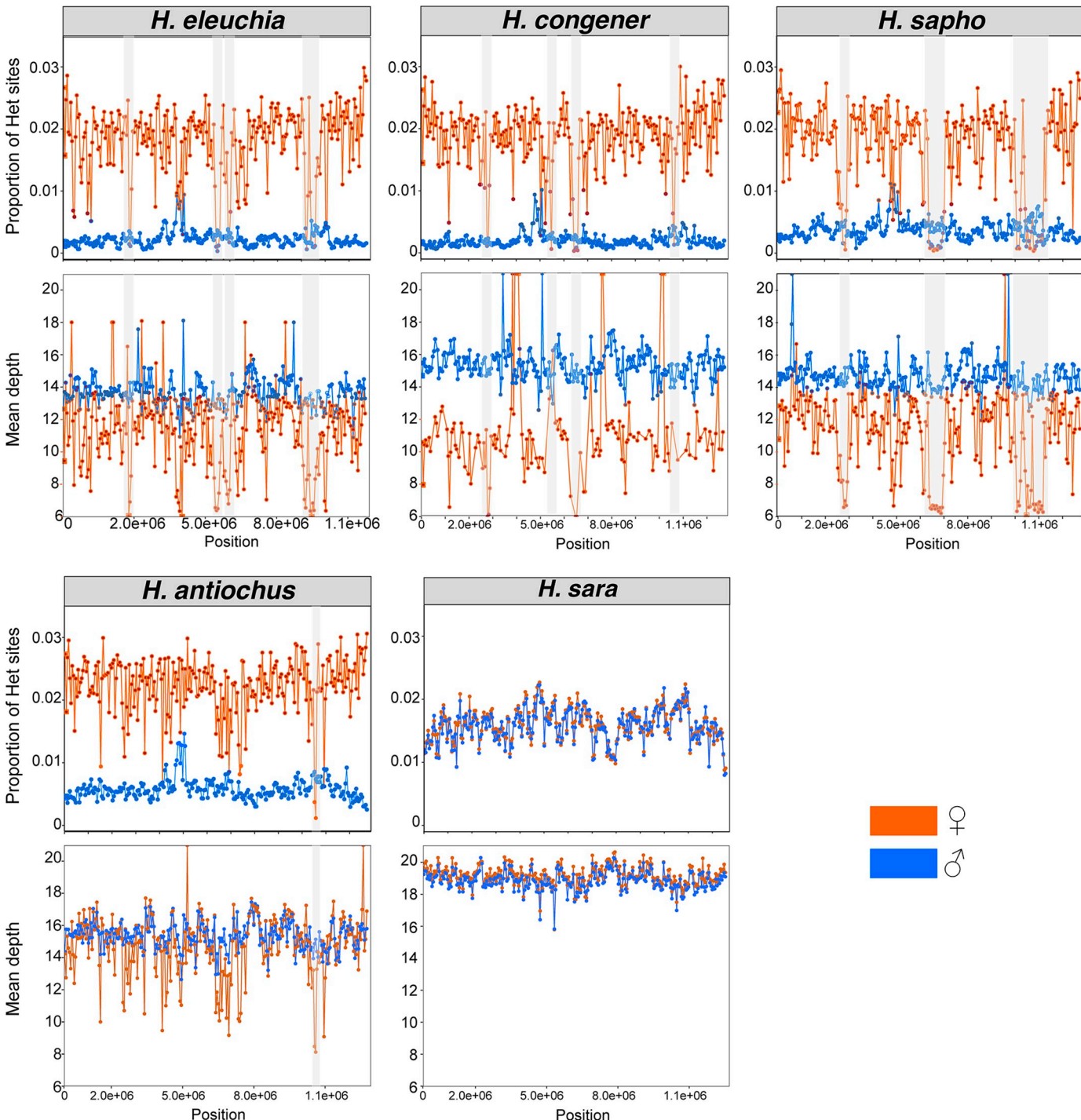

**Fig 6. Patterns of heterozygosity and depth across Chr4.** Proportion of heterozygous (Het) sites and mean sequencing depth in 50 kb sliding windows for each species. Males are shown in blue and females in orange. Grey rectangles highlight regions where females show very low sequencing depth and no heterozygous sites, suggesting that the female-limited haplotype diverged enough to become unmappable to the *H. sara* reference genome. The mean depth figures were trimmed to a maximum value of 20 to better visualize the differences between sexes.

*congener-eleuchia* subclade (Wilcoxon test, *p* <0.01) (Figs 6 and S39–S41). As expected, some peaks of high sequencing depth, likely due to expansions of repeats or duplications were also visible in both females and males (Fig 6). Interestingly, one female of *H. congener* has a region of eight windows on Chr4 with very high sequencing depth (Fig 6), indicating that repeat expansions may still be ongoing. There were also few windows on Chr4, Chr9, and Chr14 where females exhibited both lower mean sequencing depth than males and (almost) no heterozygous sites (highlighted with a grey rectangle) (Fig 6). In these regions, the female-specific haplotype likely diverged too much from the reference genome to map well and thus the heterozygosity is low in females because only one haplotype is represented in the data. *H. sara* and *H. leucadia* were the only species in the clade that did not show sex-specific patterns in heterozygosity and mean depth in Chr4, Chr9 and Chr14 (Figs 6 and S39–S41). Interestingly, the heterozygosity in females of *H. congener* and *H. sapho* dropped at the end of Chr14 and Chr9, respectively, to values similar to those of the males (S39 Fig), indicating that the last part of these chromosomes may not be fused to a sex chromosome in this species. However, the mean depth pattern did not change in these regions (S39 Fig).

## Evidence for three W-autosome fusions

In order to elucidate if the W or Z chromosome or both are involved in the sex-autosome fusions, we produced Illumina Hi-C data for a female *H. congener* (0.71 Gbp) and a male *H. sapho* (0.73 Gbp). We mapped the Hi-C data of these two *sapho* subclade individuals and the *H. sara* Hi-C data produced for the reference genome to our *H. sara* genome and to a previously published *H. charithonia* genome which has the W chromosome assembled [42]. In line with phylogenetic distances, the proportion of reads mapping to the *H. sara* reference genome were much higher (96%, 93% and 92%) than to the *H. charithonia* reference genome (69%, 70%, 71% for *H. sara*, *H. congener* and *H. sapho*, respectively). The mean mapping quality for all three species against both genomes exceeded a Phred quality score of 35.

We did not observe an excess of Hi-C contacts either between autosomes or between autosomes and sex chromosomes in the *H. sara* female (Fig 7A), as expected if the sex-autosome fusions were only present in the five species of the *sapho* subclade. The Hi-C signal of the *H. sapho* male and the *H. congener* female showed that their genomes are split into 56 and 33 chromosomes, respectively (Fig 7B and 7C), consistent with findings by Brown et al [37] and suggesting a high number of chromosome fissions. However, no fusion was observed in the *H. sapho* male suggesting that the Chr4 and Chr9 fusions are likely not with the Z, but with the W (Fig 7B). In contrast, the female *H. congener* showed an excess of contacts between Chr4 and Chr14, in line with a fusion (Fig 7C). There was no excess of Hi-C contacts between Chr4 and Chr14 with the Z chromosome (Fig 7C and 7D), indicating that the Chr4 and Chr14 are likely fused with the W instead. Even though there is some signal of Hi-C contact between these chromosomes and the W, the low mapping rates of Illumina reads of the *sapho* subclade to the W of the *Heliconius sara* genome (S2 Fig) likely explains the absence of a stronger signal. The same patterns were also observed when we mapped Hi-C data to the *H. charithonia* genome (S42 Fig).

As the Hi-C signal of chromosome fusions in the *H. congener* female represents a mix of signals from the fused and unfused haplotypes (Fig 7C and 7D), we phased the Hi-C data across the chromosomes of interest (Chr4, Chr14, W, Z) and used *Chomper* [48] to split the Hi-C data into two subsets of read pairs representing the two haplotypes (47.5% and 52.2% of the read pairs in each subset). The excess of Hi-C contacts between Chr4 and Chr14 was completely absent in one haplotype (Fig 7E) and very strong in the other (Fig 7F). The Hi-C contacts between Chr4 and Chr14 were unevenly distributed (Fig 7E) suggesting not only a

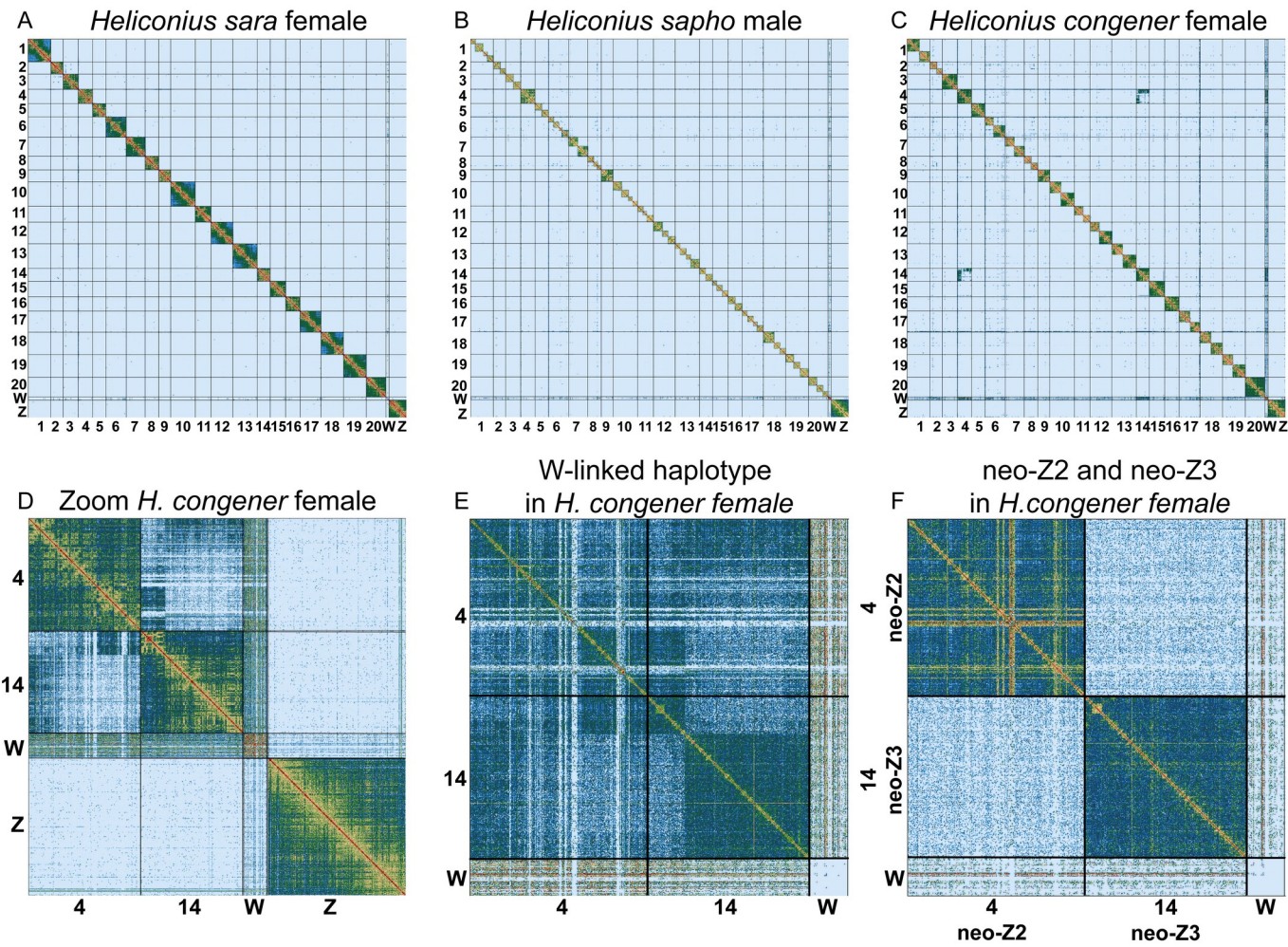

**Fig 7. Hi-C contact density map confirmed a high number of chromosome fissions and W-A fusions in the *sapho* subclade.** Hi-C contact heatmaps of the *H. sara* reference genome of (A) the *H. sara* female, (B) the *H. sapho* male, and (C) the *H. congener* female. Note that the chromosomes of the *H. sara* reference genome are labelled and their boundaries are denoted with vertical and horizontal black lines. The Hi-C signal of the *H. sapho* male and the *H. congener* female showed that their genomes are split into 56 and 33 chromosomes, respectively (B-C), consistent with the cytological chromosome counts of Brown et al [37]. (D) Zoom in (C) showing Hi-C contacts on Chr4, Chr14, ChrW, and ChrZ. (E) Hi-C reads assigned to one of the two haplotypes in the *H. congener* female showing the W-A fusion. (F) Hi-C reads assigned to the other haplotype in the *H. congener* female showing no evidence of a fusion.

fusion between these chromosomes but also the presence of other chromosomal rearrangements such as inversions or translocations.

## Discussion

We found evidence of three W-A fusions involving Chr4, Chr9, and Chr14 in the *sapho* subclade in *Heliconius* butterflies. These autosomes seem to have fused with the W chromosome as supported by: (i) females with one haplotype each forming part of female-specific haplotype clade, (ii) low $F_{ST}$ values in females, (iii) high heterozygosity and low mean depth in females, and (iv) excess of Hi-C contacts in the *H. congener* female but not in the *H. sapho* male. Because females are the heterogametic sex in butterflies and show no recombination [46], the W-A fusion is restricted to females and generates female-specific haplotypes that do not recombine with the unfused chromosomes. This results in genealogies where one haplotype of

the females clusters with male haplotypes while the other haplotype of the females, the fused one, forms a separate female-specific clade. Each female shows many heterozygous sites on the ancestral Chr4 and depending on the species also on Chr9 or Chr14, as it has inherited from the mother a fused W-4(-9/14) that never recombines and accumulates mutations and structural variants and from the father unfused Z chromosomes (Z1 = ancestral Z, Z2 = ancestral Chr4 and Z3 = ancestral Chr9 or Chr14). This, in turn, increases variation within species, but if females share the same fused haplotypes with females of other species, the variation between species is similar to the within-species variation, thus leading to low $F_{ST}$. Chr4 showed female-specific haplotype clustering and high female heterozygosity throughout the chromosome in all five species of the *sapho* subclade, indicating that the W chromosome fused with Chr4 in the ancestor of this group. In line with an old fusion and increasing degeneration, females showed a high proportion of heterozygous sites in most of the chromosome, and in some regions, the fused haplotype had degenerated so much that mapping to the *H. sara* genome failed, leading to low sequencing depth and the absence of heterozygous sites. We found similar, but less strong patterns of female-biased heterozygosity on Chr9 in the *sapho-hewitsoni* subclade, and on Chr14 in the *congener-eleuchia* subclade. In line with younger fusions and weaker degeneration, the sequencing depth of females matches that of the males on those chromosomes, indicating that the female-specific haplotypes still map well to the *H. sara* genome.

Sex-A fusions in Lepidoptera usually involve the Z chromosome [30,49], but this does not seem to be the case in *Heliconius*. Our Hi-C data for the *H. sapho* male (ZZ) showed no evidence for chromosome fusions, indicating that neither Chr4 nor Chr9 in that species is fused to the Z, whereas Hi-C data from the female *H. congener* supports a fusion on one of the haplotypes between Chr4 and Chr14 with the W chromosome. Thus, in *H. congener*, the unfused Chr4 and Chr14 become neo-Z2 and neo-Z3 chromosomes, respectively, and the fused chromosome becomes the neo-W chromosome. Although the absence of a fusion with the Z in the female *H. congener* clearly points to a W fusion, Hi-C contacts between Chr4-Chr14 and the W are weak. This is likely due to the divergence of the W in the *sapho* subclade from that of *H. sara*, making it difficult to map reads against this reference (S2 Fig). Alternatively, the original W in the *sapho* subclade may have an independent origin to that in *H. sara* or it may have been lost completely, as suggested for other Lepidoptera [49].

Even though, we only generated Hi-C data for one representative each of the *congener-eleuchia* subclade and the *sapho-hewitsoni* subclade, we conclude that all sex-autosome fusions likely involve the W chromosome, as the marginal phylogenies showed that the Chr4-fusion is ancestral to all five species of the *sapho* subclade (Fig 3), and the subsequent Chr9 or Chr14 fusions are ancestral to the *sapho-hewitsoni* subclade (S29 Fig) or *congener-eleuchia* subclade (S30 Fig), respectively. However, we cannot exclude the possibility that the ancestral Z chromosome additionally fused with the neo-Z chromosomes (former Chr4/Chr9/Chr14) in some of the species.

The finding of multiple W-A fusions in the *sapho* subclade is particularly striking since this group is known for its high number of chromosomes compared to all other *Heliconius* species that was also confirmed with our Hi-C data of two species. While most species in this genus have 21 chromosomes, *H. sapho* and *H. eleuchia eleusinus* and *H. e. primularis* has 56–57 chromosomes, and *H. eleuchia eleuchia* and *H. congener* have 37 and 33 chromosomes, respectively [37], indicative of high rates of chromosomal fission events in the group. However, even though in these species most autosomes are broken up (Fig 7B and 7C), the chromosomes fused with the W show no signs of fissions. Interestingly, *H. antiochus* and *H. hewitsoni* which also show W-A fusions have 21 chromosomes. These findings could be explained by two alternative scenarios: (i) the chromosomal fissions in *H. eleuchia*, *H. congener* and *H. sapho* happened independently in each (sub)species after the W-A fusions and fusing to the W protected

Chr4 and Chr9/14 from fissions, or (ii) the fissions could be ancestral, and the W-fusions could have involved the largest chromosomes that had not been broken up. In this second scenario, *H. hewitsoni* would have undergone additional autosomal fusion events restoring chromosome number to 21 after fission events. While we think the first scenario is more likely than the second, full genome assemblies will be required to distinguish these hypotheses.

While we cannot test for an adaptive role of the W-A fusions in the *sapho* subclade, the fact that they occurred three times and remained fixed in multiple species suggests they are at least not deleterious, or that any deleterious effect is masked. In the latter scenario, low recombination around the W-A fusions would force their sex-specific transmission and result in a permanent heterozygosity that protects against the expression of deleterious recessive mutations load and favours the accumulation of adaptive mutations. This is the case in inversions in *H. numata* [3,50]. Other alternative scenarios may have favoured the fixation of these fusions. First, a period of strong genetic drift could have led to the fixation of these fusions even if they carry some deleterious mutations. Second, positive natural selection acting on fusions is possible [51], though it might be hard to imagine that such beneficial effects would be found on all three chromosomes that fused to the W. Third, transmission bias such as meiotic drive [52,53], or coincidental linkage with endosymbionts transmitted via females (e.g. male-killing spiroplasma) [54] could explain how the W-A fusions might have fixed. However, if chromosomal fusions occur through non-homologous recombination, meiotic drive might counteract or facilitate the spread of the fusions as observed for *Leptidea* butterflies [55]. Fourth, the W-A fusions might have spread due to reduction of sexual conflict if there are sexually antagonistic loci on Chr4/9/14. In Danaini butterflies, sex-biased gene expression is consistent with this hypothesis [11]. Finally, the W-A fusions may have contributed to the particularly high diversification rate in this clade if they linked together barrier loci in regions with reduced recombination [14].

This is the first genomic study focused on the *sara/sapho* clade. The inclusion of multiple species and subspecies of this clade from a broad geographic range also allowed us to redefine some of the relations previously reported [38], and to identify the effect of geography in shaping diversity. The phylogenetic position we found for *H. antiochus* and *H. hewitsoni* contrasts with previous amplicon based phylogenies [38,56] but agrees with a recent whole genome phylogeny based on *de novo* genome assemblies [57], suggesting that the phylogenetic relations we describe for these two species are the most plausible. We also identified *cis* and *trans*-Andean lineages for *H. sara* and *H. antiochus*, as well as *H. congener* and *H. eleuchia* structured by the Andes (Fig 1).

Further studies are needed to understand the evolutionary drivers of the W-A fusions identified here, as well as their role (if any) in speciation or adaptation in this clade. Our study highlights the importance of including both sexes in short-read population WGS studies for identifying Sex-A fusions. Finally, we show what patterns to expect if the fusions are recent enough that the previously autosomal chromosomes are still diploid in both sexes. As W chromosomes are often not assembled in reference genomes due to their high repeat content, genome assemblies might not necessarily reveal W-A fusions. Our study thus demonstrates the power of short-read population data to detect the genomic signatures left by sex-A fusions, particularly for taxa where one sex is achiasmic.

## Materials and methods

### Genome assembly of *Heliconius sara magdalena*

We used two laboratory-reared females from a stock population from Panama to generate a reference genome for *H. sara magdalena* (BioSamples SAMEA8947140 and SAMEA8947139;

S1 Table). We assembled the genome by combining PacBio, 10X data and Hi-C data, all generated by the Tree of Life Programme at the Wellcome Sanger Institute (https://www.sanger.ac.uk/programme/tree-of-life/). The BioSample SAMEA8947140 was used to generate the PacBio continuous long reads (CLR). Libraries were sequenced on four Single Molecule Real-Time (SMRT) cells using the PacBio Sequel II system. The linked-reads from 10X Genomics Chromium technology were generated with the same sample and sequenced in four lanes on the Illumina HiSeq X Ten platform. The second BioSample SAMEA8947139 was used to produce Dovetail Hi-C data and sequenced on a HiSeq X Ten platform.

An initial contig assembly was generated from the PacBio (CLR) data using wtdbg2 v2.2 [58]. The PacBio data was then used to polish the contigs using Arrow (https://github.com/PacificBiosciences/GenomicConsensus). We then retained haplotig identification with the Purge Haplotigs pipeline [59]. The 10X data were mapped to this assembly using Longranger v2.2 (10X Genomics) and variant calling was performed using freebayes v1.1.0-3-g961e5f3 [60]. Next, this first assembly was polished using BCFtools consensus v1.9 [61] by applying homozygous non-reference calls as edits. The 10X linked-reads were then used to scaffold contigs using Scaff10X v2.3 (https://github.com/wtsi-hpag/Scaff10X). A round of manual curation was performed on these polished scaffolds using gEVAL [62]. Lastly, Dovetail Genomics Hi-C data was used to scaffold the assembly further using SALSA v2.2 [63], followed by another round of manual curation with gEVAL [62]. The chromosome-scale scaffolds were named by synteny to the *Heliconius melpomene melpomene* assembly Hmel2.5 in LepBase. We assessed the genome contiguity with gnx-tools (https://github.com/mh11/gnx-tools/blob/master/README) and genome completeness with BUSCO v5.1.2 [64] using the Lepidoptera gene set. To obtain synteny plots between *H. sara* vs. *H. melpomene*, *H. erato*, and *H. charithonia*, we first performed pairwise alignments between these genomes using minimap2 v. 2.24 [65]. Subsequently, we plotted the minimap2 results using custom scripts from (https://github.com/simonhmartin/asynt).

We used whole genome resequencing data from 114 individuals obtained in this study (see sample collection for genome resequencing section) to identify the W chromosome within the genome of *Heliconius sara*. We calculated the mean depth across the scaffolds that were not yet assigned to a chromosome. We first generated a file containing the mean depth per site averaged across all individuals of the same sex and species using the—site-mean-depth option of vcftools v.0.1.14 [66]. Then, we used the R package windowscanr v. 0.1 (https://github.com/tavareshugo/WindowScanR) to calculate the average of the mean depth per species, per sex, and per 500 bp windows. Scaffolds where we observed a higher mean depth in females compared to males were assigned to the W chromosome.

## Sample collection for genome resequencing

We collected 114 *Heliconius* individuals from 7 species and 18 subspecies in the *sara/sapho* clade across their distribution range: 48 *H. sara*, 2 *H. leucadia*, 21 *H. antiochus*, 13 *H. sapho*, 3 *H. hewitsoni*, 17 *H. eleuchia* and 10 *H. congener* (S1 Table). The body of each individual was preserved in NaCl-saturated DMSO solution and stored at -80°C; wings were kept for phenotype reference.

## Whole-genome resequencing and genotype calling

Genomic DNA was extracted from thoracic tissue using a DNeasy Blood and Tissue Kit (Qiagen). Library preparation and whole-genome Illumina resequencing (PE reads) was carried out on Illumina's HiSeq X system by Novogene (Beijing, China), with 30X coverage per individual. We also downloaded two samples of *H. charithonia* (SRR4032025 –SRR4032026) from SRA

(https://www.ncbi.nlm.nih.gov/sra) to include them as outgroups in phylogenetic analyses. Our *H. sara* genome (HelSar1) was used as a reference to map the reads of each individual using BWA mem v0.7.12 [67] with default parameters. We then used samtools v1.12 to sort and index the alignment files [68]. PCR-duplicate reads were identified and removed using Picard tools v2.9.2 [69], and variant calling was conducted with HaplotypeCaller (GATK, v3.7.0) in BP-resolution mode [70]. Then, samples were jointly genotyped using GATK's GenotypeGVCFs (68). We used vcftools v0.1.14 [66] and the final VCF to calculate: (i) mean depth per individual and site, (ii) quality per site, (iii) the proportion of missing data per individual and (iv) the proportion of missing data per site, and (v) mapping percentage per individual. Based on these results, we kept sites with quality value (—minQ) $\geq$ 30 and less than 5% missing data. We also excluded sites with a sequencing depth below 5 and mean depth per individual more than 1.5 times the mean to exclude paralogous regions. For this, we used the custom script *removeTooHighDepthSites.sh* from (https://github.com/joanam/VictoriaRegionSuperflock/BashPipelines). We additionally removed sites with excess heterozygosity across all individuals using the vcftools option—*hardy* and a p-value cut-off of <1e-5 to remove reads from paralogous regions that are collapsed in the reference genome.

## Analysis of population structure within the *sara/sapho* clade

We performed a principal component analysis (PCA) to study the genetic structure of populations. We filtered out monomorphic or multiallelic sites, and sites with minor allele frequency (MAF) smaller than 0.1 with vcftools [66]. To reduce the linkage disequilibrium effect, we used the python script *ldPruning.sh* from (https://github.com/joanam/scripts), which removes sites with $r^2$>0.2 in windows of 50 Kbp sliding by 10 Kbp. This resulted in a vcf file with 3,685,916 high-quality SNPs sites. We conducted the PCA using Plink v2.0 with default parameters [71,72].

## Phylogenetic relationships among *sara/sapho* clade species

We generated a whole-genome Maximum Likelihood (ML) tree using a vcf containing all sites as input in RAXML v8.2.9 [73], with the GTRGAMMA model and 100 bootstrap replicates. We applied the same procedure to obtain ML trees for each chromosome to study the phylogenetic incongruence across the genome. We also inferred the species tree using the coalescent-based method ASTRAL [74]. For this, we used vcftools v0.1.14 [66] to: (i) extract two males per subspecies so Sex-A fusions present only in females do not alter the species tree, and (ii) extract 2 kbp loci spaced at least 10 kbp apart to ensure no linkage disequilibrium between them [75]. Then, samtools [68] was used to generate 271 multilocus blocks, each resulting from concatenating 100 loci. Each block was converted into PHYLIP format using our custom script *vcf2phylip.py* from (https://github.com/joanam/scripts/blob/master/vcf2phylip.py) and used to estimate a ML tree in IQ-tree selecting the best model with ModelFinder and assessing node support with 1000 ultrafast bootstraps [76,77]. The resulting 271 topologies were used as input in ASTRAL. We also investigated these 271 topologies with DensiTree [78] to visualize discordance.

## Haplotype-based phylogenetic analysis on chromosomes 4, 9 and 14

To further investigate sex clustering in the phylogenies of autosomes 4, 9 and 14, we phased them to infer their haplotypes. Haplotype phasing was done by combining two methods: WhatsHap, which is a haplotype assembly technique [79], and SHAPEIT4, a statistical phasing method [80]. To implement WhatsHap, we used the BAM file of each individual as well as the reference genome to group nearby genetic variants into fully resolved haplotype blocks or

phase sets [79]. Then, we used the WhatsHap output file to run SHAPEIT4, which further phases haplotypes based on population-level information using default parameters [80]. Despite combining these different phasing approaches, our dataset still included phasing errors leading to haplotype switching in the females between the W-linked chromosomes and the neo-Z chromosomes. We were thus not able to generate phylogenies of the complete chromosomes and instead used Relate v.1.1.2, a method that uses short-range phasing information to infer phylogenies.

With Relate v.1.1.2 [47] we inferred ancestral recombination graphs (ARG) along chromosomes 4, 9, and 14. This software generates genealogies by first identifying the relative order of coalescence events at each genomic position using a nonsymmetric distance matrix and considering a mutation rate and a recombination map. This matrix is constructed based on posterior probabilities from a hidden Markov model (HMM), which considers ancestral and derived SNP status to enhance accuracy. Subsequently, a rooted binary tree is constructed from this distance matrix [47]. Initially, the phased vcf file was transformed into haplotype format using RelateFileFormats with the "—mode ConvertFromVcf" flag. We performed this analysis using an effective population size of $1x10^7$ individuals, as estimated for *H. erato* [81], and a mutation rate of $2.9x10^{-9}$ per site per generation from *H. melpomene* [82]. As Relate further requires a genetic map, we created a uniform recombination rate map using the average recombination rate of 6 cM/Mb calculated in *H. erato* [83], and our custom script (https://github.com/joanam/scripts/blob/master/createuniformrecmap.r). Finally, the ancestral allele was assigned as the one more common in the outgroup *H. charithonia*.

Because any possible Sex-A fusion would produce heterozygous females and homozygous males, we identified sites with this pattern (see hypotheses in Fig 2). We first calculated the number of heterozygous and homozygous individuals per species and site using the—*hardy* option in vcftools v0.1.14 (46). Then, we used custom scripts to find sites where each species in the *sapho* subclade met the following criteria: (i) no males were heterozygous, (ii) at least one female was heterozygous, and (iii) not more than one female was homozygous (allowing for one female with one allele not called). Finally, we selected one SNP for every 1000 SNPs from these filtered sites and visualized their genealogies using the script *TreeView.sh* (58). This subsampling approach enabled us to examine genealogies from various sites evenly distributed along the chromosomes.

## Patterns of genetic differentiation

We calculated $F_{ST}$ by pairs of sister species along chromosomes in non-overlapping 50 Kbp windows. Because *H. antiochus* did not have a sister species, we calculated these statistics between Andean and Amazonian subspecies. Windows that contained less than 2,500 high-quality genotyped variable sites were rejected. We used a dataset including SNPs and monomorphic sites and the *popgenWindows.py* script from (https://github.com/simonhmartin/genomics_general). We also calculated $F_{ST}$ per sex following the same methodology.

## Patterns of heterozygosity and mean depth by chromosome

To study chromosomes with $F_{ST}$ patterns different from the genome average, we used the options—het and—depth of vcftools v.0.1.14 [66] to calculate heterozygosity and mean depth per chromosome for each individual of each species. We also calculated these statistics in 50 Kbp non-overlapping sliding windows along the 'outlier' chromosomes identified. On these, we calculated π specifying each individual as its own population, so π became a measure of proportion of heterozygous sites. This was done with the Python script *popgenWindows.py* from (https://github.com/simonhmartin/genomics_general). We then averaged these values

across all individuals of the same sex and species. For sequencing depth, we first generated a file containing the mean depth per site averaged across all individuals of the same sex and species using the—site-mean-depth option of vcftools v. 0.1.14 [66]. We then used the R package windowscanr v.0.1 from (https://github.com/tavareshugo/WindowScanR) to calculate the mean of the mean depth per species, per sex, and per window. The few individuals of *H. hewitsoni* and *H. leucadia* were not included in the sliding windows analysis. Statistical tests were applied to assess significant differences in heterozygosity and mean depth between sexes and between chromosomes. As the data were not normally distributed, we performed a Wilcoxon signed rank test to compare the sexes. To assess differences between chromosomes, we applied a Kruskal-Wallis test and a *post hoc* test (pairwise Wilcoxon test for Kruskal-Wallis).

## Identification of fusions with Hi-C data

To investigate whether autosomes 4, 9, and 14 are fused with the W or Z chromosomes, we constructed Hi-C libraries from the thorax of one female of *Heliconius congener* (BioSamples SAMEA112329098; S1 Table) and a male of *Heliconius sapho* (BioSamples SAMEA112696452, S1 Table) using the Arima2 kit (Arima Genomics, Inc). These libraries were then sequenced on an Illumina NovaSeq S4 platform with 150 bp paired-end reads. We used BWA mem2 v 2.2.1 [84] to map the reads against the genomes of *Heliconius sara* (this study) and *Heliconius charithonia* [42] using default parameters. We also mapped the Hi-C data of *H. sara* to both references to confirm the absence of the fusion in the *sara* subclade. Next, we removed PCR duplicates, eliminated poorly aligned reads, and filtered out reads with a mapping quality <10 from the resulting BAM files using samtools v1.12 [68]. Additionally, we generated contact maps with pretextview and pretextSnapshot (https://github.com/wtsi-hpag/PretextView and https://github.com/wtsi-hpag/PretextSnapshot). All these steps were done using a custom Perl pipeline developed by Shane McCarthy at the Wellcome Sanger Institute.

To further investigate the excess of Hi-C contacts between autosomes 4, 14 and W in the female of *H. congener*, we generated haplotype-specific Hi-C maps for these chromosomes. First, we created a version of the *H. sara* reference genome in which chromosomes 4, 14, and W were concatenated together to allow for phasing across chromosomes. Next, we mapped the Hi-C reads of *H. congener* to this modified reference following the mapping pipeline by Arima Genomics, Inc. (https://github.com/ArimaGenomics/mapping_pipeline). Then we called heterozygous variants using freebayes v1.3.2-dirty [60]. These variants were then normalized with bcftools v1.8 [61], decomposed with vcfallelicprimitives [85] and filtered for coverage (>21 and <141 reads) with vcftools v. 0.1.14 [66]. Next, the remaining SNPs were phased using HAPCUT2 v1.3.3, using both the bam and the vcf files as input [86]. We used the Python script *chomper.py* from to separate haplotype aligned Hi-C reads [48]. Finally, these haplotype-specific sets of Hi-C reads were realigned to the original *H. sara* assembly using a custom Perl pipeline developed by Shane McCarthy at the Wellcome Sanger Institute. Hi-C contact map was generated using pretextview and pretextSnapshot (https://github.com/wtsi-hpag/PretextView and https://github.com/wtsi-hpag/PretextSnapshot).

## Supporting information

**S1 Table. Sample information and genotyping statistics.** Individuals in bold were not included in the analyses due to low depth and missing data. *These specimens were used to generate the reference genome of *Heliconius sara*. **These specimens were used to generate the HiC data.
(XLSX)

**S2 Table. Lepidoptera genome assembly statistics modified from [81].** We added our new *H. sara* assembly and the *H. charithonia* assembly from [42].
(XLSX)

**S3 Table. BUSCO results statistics modified from [81].** We added our *H. sara* assembly and the *H. charithonia* assembly of [42].
(XLSX)

**S1 Fig. Synteny plots showing high collinearity between *Heliconius* genomes.** Pairwise alignment between chromosomes of (A) *H. sara* and *H. erato*, (B) *H. sara* and *H. melpomene*, and (C) *H. sara* and *H. charithonia*. The GenBank accession numbers for the genomes of *H. melpomene*, *H. erato* and *H. charithonia* are GCA_000313835.2, GCA_018249695.1 and GCA_030704555.1, respectively. The W chromosome in *H. sara* corresponds to a single homolog in H. *charithonia*. Because the W chromosome was not assembled in the genomes of *H. erato* and *H. melpomene* genomes, we could not compare the W of *H. sara* W against them.
(TIF)

**S2 Fig. Identification of the W chromosome in the genome of *H. sara*.** (A) Genome-wide topology of the *sara-sapho* clade. (B) Mean depth vs. position along the scaffold in each species. We plotted one (scaffold 81) out of the 32 scaffolds where females of the species *H. sara* showed half the sequencing depth of autosomes and where males do not map. The mean depth of an autosomal chromosome (Chr4) of the species *H. sara* is provided as an example for comparison with scaffold 81. Females are shown in red and males in blue.
(TIF)

**S3 Fig. Missing data and mean depth per individual.** Each species is symbolised by a unique symbol and colour. Note that the lower missing data proportion in *H. sara* is likely due to its similarity with the reference genome (*H. sara* female).
(TIF)

**S4 Fig. Principal Component Analysis (PCA), performed with 3,685,916 SNPs.** The PCA groups the individuals into two main groups: (i) *H. sara and leucadia* (hereafter *sara* subclade), and (ii) *H. antiochus*, *H. eleuchia*, *H. congener*, *H. sapho* and *H. hewitsoni* (hereafter *sapho* subclade). The first two principal components explain 60% (PC1) and 10% (PC2) of the total variance, respectively. PC1 separates the *sara* subclade from the *sapho* subclade, whereas PC2 separates *H. antiochus* from the rest of the species of the *sapho* subclade. *H. sapho* is closer to *H. hewitsoni*, whereas *H. eleuchia* is closer to *H. congener*.
(TIF)

**S5 Fig. Species tree based on ASTRAL multi-species coalescence and Densitree.** (A) ASTRAL species tree based on 271 phylogenetic trees (each recovered from a block of 100 loci). Branch lengths are shown in coalescent units. All nodes are supported with a bootstrap value of 100%. (B) DensiTree calculated from 271 topologies showing phylogenetic discordance within the *sara-sapho* clade.
(TIF)

**S6 Fig. Maximum likelihood phylogeny of chromosome 1.** Bootstrap support values are indicated at branches.
(TIF)

**S7 Fig. Maximum likelihood phylogeny of chromosome 2.** Bootstrap support values are indicated at branches.
(TIF)

**S8 Fig. Maximum likelihood phylogeny of chromosome 3.** Bootstrap support values are indicated at branches.
(TIF)

**S9 Fig. Maximum likelihood phylogeny of chromosome 4.** Bootstrap support values are indicated at branches. Blue squares group males and while red squares group females.
(TIF)

**S10 Fig. Maximum likelihood phylogeny of chromosome 5.** Bootstrap support values are indicated at branches.
(TIF)

**S11 Fig. Maximum likelihood phylogeny of chromosome 6.** Bootstrap support values are indicated at branches.
(TIF)

**S12 Fig. Maximum likelihood phylogeny of chromosome 7.** Bootstrap support values are indicated at branches.
(TIF)

**S13 Fig. Maximum likelihood phylogeny of chromosome 8.** Bootstrap support values are indicated at branches.
(TIF)

**S14 Fig. Maximum likelihood phylogeny of chromosome 9.** Bootstrap support values are indicated at branches. Blue squares group males and while red squares group females.
(TIF)

**S15 Fig. Maximum likelihood phylogeny of chromosome 10.** Bootstrap support values are indicated at branches.
(TIF)

**S16 Fig. Maximum likelihood phylogeny of chromosome 11.** Bootstrap support values are indicated at branches.
(TIF)

**S17 Fig. Maximum likelihood phylogeny of chromosome 12.** Bootstrap support values are indicated at branches.
(TIF)

**S18 Fig. Maximum likelihood phylogeny of chromosome 13.** Bootstrap support values are indicated at branches.
(TIF)

**S19 Fig. Maximum likelihood phylogeny of chromosome 14.** Bootstrap support values are indicated at branches. Blue squares group males and while red squares group females.
(TIF)

**S20 Fig. Maximum likelihood phylogeny of chromosome 15.** Bootstrap support values are indicated at branches.
(TIF)

**S21 Fig. Maximum likelihood phylogeny of chromosome 16.** Bootstrap support values are indicated at branches.
(TIF)

**S22 Fig. Maximum likelihood phylogeny of chromosome 17.** Bootstrap support values are indicated at branches.
(TIF)

**S23 Fig. Maximum likelihood phylogeny of chromosome 18 of the Sara-Sapho clade.** Bootstrap support values are indicated at branches, and the scale bar represents the percentage of substitutions per site.
(TIF)

**S24 Fig. Maximum likelihood phylogeny of chromosome 19.** Bootstrap support values are indicated at branches.
(TIF)

**S25 Fig. Maximum likelihood phylogeny of chromosome 20.** Bootstrap support values are indicated at branches.
(TIF)

**S26 Fig. Maximum likelihood phylogeny of chromosome Z.** Bootstrap support values are indicated at branches.
(TIF)

**S27 Fig. Maximum Likelihood (ML) phylogenies inferred genome-wide and per chromosome.** (A) Topologies found across the genome. Purple: genome-wide topology. Green: *H. congener* within *H. eleuchia*. Orange: *H. hewitsoni* as sister to *H. congener + H. eleuchia*. Brown: *H. hewitsoni* as sister to *H. congener + H. eleuchia + H. sapho*. Chromosomes are shown in the bottom, with coloured triangles indicating the topology revealed by each of them. (B) Topologies showing sex-specific grouping within some species, which is indicative of Sex-A fusions in Chr4, Chr9 and Chr14. In these species, females are coloured in red and males in blue.
(TIF)

**S28 Fig. Marginal tree for one SNP extracted from an ancestral recombination graph of Chr4.** (A) This is one of the 112 marginal phylogenies where female alleles from at least two species in the sapho subclade cluster as expected in a Sex-A fusion. Each vertical line represents an individual allele, and the alleles of all individuals are shown differentiating those of females (orange) from those of males (blue). Note that one allele of the females clusters with the alleles of males, while the other female allele formed a separate group (highlighted in orange).
(TIF)

**S29 Fig. Marginal tree for haplotypes around a focal SNP extracted from an ancestral recombination graph of Chr9.** (A) This is one of the 15 haplotypes genealogies that showed a consistent pattern with the W-Sex fusion. The vertical lines represent the alleles of each SNP for each individual. Each vertical line represents an individual haplotype, and the alleles of all individuals are shown differentiating those of females (orange) from those of males (blue). Note that one haplotype of the females clusters with the alleles of males, while the other female allele formed a separate group (highlighted in orange). B) Position of each of the 15 SNPs in Chr9 that show genealogies consistent with a W-Sex fusion; they were not clustered in a specific region but rather distributed along the entire chromosome. The position of the SNP

whose genealogy shown in A, is indicated by an arrow.
(TIF)

**S30 Fig. Marginal tree for haplotypes around a focal SNP extracted from an ancestral recombination graph of Chr14.** (A) This is one of the 25 haplotypes genealogies that showed a consistent pattern with the W-Sex fusion. Each vertical line represents an individual allele, and the alleles of all individuals are shown differentiating those of females (orange) from those of males (blue). Note that one allele of the females clusters with the alleles of males, while the other female allele formed a separate group (highlighted in orange). B) Position of each of the 25 SNPs in Chr14 that show genealogies consistent with a W-Sex fusion; they were not clustered in a specific region but rather distributed along the entire chromosome. The position of the SNP whose genealogy shown in A, is indicated by an arrow.
(TIF)

**S31 Fig. Genome-wide divergence ($F_{ST}$) between pairs of species in the *sara/sapho* clade.** Each dot represents a chromosome, and chromosomes with evidence of Sex-A fusions are colour coded.
(TIF)

**S32 Fig. Genetic divergence ($D_{XY}$) and population nucleotide diversity ($\pi$) in the sara/ sapho clade.** (A) $D_{XY}$ between pairs of species. Each point represents a 50Kb window, whereby the. the top 5% windows are shown in black. The numbers below correspond to *H. sara* chromosomes. (B) $\pi$ per species and by sex. Each circle represents a chromosome, and chromosomes with evidence of Sex-A fusions are colour coded.
(TIF)

**S33 Fig. Genome-wide divergence ($F_{ST}$) between pairs of subspecies of *H. eleuchia*, *H. congener* and *H. sapho*.** Each point represents a 50Kb window. The significance threshold is set at the top 5% of the $F_{ST}$ values distribution tail, and black windows are those that passed this threshold.
(TIF)

**S34 Fig. Genome-wide divergence ($F_{ST}$) between pairs of subspecies of *H. antiochus*.** Each point represents a 50Kb window. The significance threshold is set at the top 5% of the $F_{ST}$ values distribution tail, and black windows are those that passed this threshold.
(TIF)

**S35 Fig. Genome-wide divergence ($F_{ST}$) between pairs of subspecies of *H. sara*.** (a) $F_{ST.}$ Each point represents a 50Kb window. The significance threshold is set at the top 5% of the $F_{ST}$ values distribution tail, and black windows are those that passed this threshold.
(TIF)

**S36 Fig. Genome-wide divergence ($F_{ST}$) in the *sara/sapho* clade.** $F_{ST}$ between (A) subspecies of *H. antiochus*, (B) *H. congener* and *H. eleuchia*, (C) *H. sapho* and *H. hewitsoni*, and (D) *H. sara* and *H. leucadia*. $F_{ST}$ was calculated by sex and are colour coded. **$p < 0.05$.
(TIF)

**S37 Fig. Patterns of heterozygosity across the genome in the *sara/sapho* clade.** Proportion of heterozygous sites by sex in each species: (A) *H. antiochus*, (B) *H. congener*, (C) *H. eleuchia*, (D) *H. sapho*, and (E) *H. sara*. Sexes are colour coded. **$p < 0.05$.
(TIF)

**S38 Fig. Patterns of mean depth across the genome in the *sara/sapho* clade.** Mean depth by sex in each species: (A) *H. antiochus*, (B) *H. congener*, (C) *H. eleuchia*, (D) *H. sapho*, and (E) *H. sara*. Sexes are colour coded. **p<0.05.
(TIF)

**S39 Fig. Patterns of heterozygosity and depth across chromosome (A) 14 and (B) 9.** Proportion of heterozygosity sites and mean depth in 50Kb sliding windows in each species. Each line corresponds to one individual where males are shown in blue and females in red, and their *n* is shown in the top right corner. In all species, the sex-linked chromosomes show even patterns of increased heterozygosity, except for the end of Chr14 in *H. congener*.
(TIF)

**S40 Fig. Proportion of heterozygous sites and mean depth between sexes in sliding windows along chromosome 4.** Each panel corresponds to a species with the proportion of heterozygous sites shown at the top and mean depth at the bottom. Each dot represents the average of these values across all individuals per window. ns = non-significant.
(TIF)

**S41 Fig. Proportion of heterozygous sites and mean depth between sexes in sliding windows along chromosome (A) 14 and (B) 9.** Each panel corresponds to a species with the proportion of heterozygous sites shown at the top and mean depth at the bottom. Each dot represents the average of these values across all individuals per window. ns = non-significant.
(TIF)

**S42 Fig. The density map of Hi-C contacts confirms W-A fusions in the sapho subclade.** Hi-C contact heatmaps of the genome assembly of *H. charithonia* against: (A) the *H. sara* female (Hi-C used to assemble the genome reported here), (B) the *H. sapho* male, and (C) the female of *H. congener*
(TIF)

## Acknowledgments

We thank "Autoridad Nacional de Licencias Ambientales—ANLA" in Colombia for granting Universidad del Rosario the collecting permit 530 and the Instituto Chico Mendes de Conservação da Biodiversidade in Brazil for granting SISBIO collection licence 59194–1 under which we performed our collecting activities. We also thank the HPC Service of Universidad del Rosario (CALDAS) for computing time. We thank the core lab of the Tree of Life Programme and the sequencing centre of the Sanger Institute for support with the Hi-C sequencing.

## Author Contributions

**Conceptualization:** Nicol Rueda-M, Joana I. Meier, Camilo Salazar.

**Data curation:** Nicol Rueda-M.

**Formal analysis:** Nicol Rueda-M, Gabriela Montejo-Kovacevich, Shane McCarthy, Joana I. Meier, Camilo Salazar.

**Funding acquisition:** Nicol Rueda-M, Camilo Salazar.

**Investigation:** Nicol Rueda-M, Chris D. Jiggins, Camilo Salazar.

**Methodology:** Nicol Rueda-M, Joana I. Meier, Camilo Salazar.

**Resources:** Nicol Rueda-M, Carolina Pardo-Diaz, W. Owen McMillan, Krzysztof M. Kozak, Carlos F. Arias, Jonathan Ready, Richard Durbin, Chris D. Jiggins, Camilo Salazar.

**Supervision:** Joana I. Meier, Camilo Salazar.

**Visualization:** Nicol Rueda-M, Carolina Pardo-Diaz, Joana I. Meier, Camilo Salazar.

**Writing – original draft:** Nicol Rueda-M, Carolina Pardo-Diaz, Joana I. Meier, Camilo Salazar.

**Writing – review & editing:** Nicol Rueda-M, Carolina Pardo-Diaz, Gabriela Montejo-Kovacevich, W. Owen McMillan, Krzysztof M. Kozak, Carlos F. Arias, Jonathan Ready, Shane McCarthy, Richard Durbin, Chris D. Jiggins, Joana I. Meier, Camilo Salazar.

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
