## [Decision Letter · Decision Letter 0]

23 May 2023

Dear Dr Salazar,

Thank you very much for submitting your Research Article entitled 'Three sequential sex chromosome – autosome fusions in Heliconius butterflies' to PLOS Genetics.

The manuscript was fully evaluated at the editorial level and by independent peer reviewers. The reviewers appreciated the attention to an important problem, but raised some substantial concerns about the current manuscript. Based on the reviews, we will not be able to accept this version of the manuscript, but we would be willing to review a much-revised version. We cannot, of course, promise publication at that time.

If you decide to revise the manuscript for further consideration at PLOS Genetics, please aim to resubmit within the next 60 days, unless it will take extra time to address the concerns of the reviewers, in which case we would appreciate an expected resubmission date by email to plosgenetics@plos.org.

We are sorry that we cannot be more positive about your manuscript at this stage. Please do not hesitate to contact us if you have any concerns or questions.

Yours sincerely,

James Mallet

Guest Editor

PLOS Genetics

Gregory Barsh

Editor-in-Chief

PLOS Genetics

This is an interesting paper concerning the genomics of a phylogenetic sub-group of Heliconius butterflies that has hitherto not been tackled. I have good evidence that the clade was in fact deliberately ignored for genome sequencing to avoid assembly and interpretation complications due to the known rampant chromosomal evolution previously demonstrated by testes-based chromosomal counts. Almost all Heliconius are highly syntenic and have n = 21 chromosomes, but the sara-sapho group may have n = 21-60, and are sometimes apparently polymorphic for chromosome number within species. And the expectation of interpretation difficulties has now been amply demonstrated by the current authors! The current work provides the first genomic analysis of that clade, and is based on a high quality chromosomal scale reference genome for H. sara (n = 21) as well as reasonably high coverage Illumina resequence data for all the other species in the clade and many of their subspecies.

Reviewer 1 recommends rejection; Reviewer 2 recommends minor revision, and Reviewer 3 major revision. I am therefore inclined to suggest major revision, but I do think that the very important critiques of Reviewer 1 need to be answered. My own feelings about the manuscript were disappointment that the paper was not more definitive. The authors have some strong evidence for genomic shenanigans, but I agree with the reviewers that they have only indirect evidence to support their claims about fusion of autosomes with sex-chromosomes in the sapho-eleuchia subclade. This is because no assemblies were done of the putatively fused species, and so the “phylogenetic” pattern of males and females appearing on separate branches, strange heterozygosity differences in females, and patterns of divergence are the only evidence of sex-autosome chromosomal fusions.

The reviewers do not dwell on this, but I was rather shocked that there is not more discussion of the high quality reference genome of H. sara. Instead, all we get is a 6-line paragraph with the title “Genome assembly” in the results. I infer that the genome assembled is H. sara, but the species is not even mentioned in the Results! There are no genome coverage, or other statistics that I could find (except some mention of BUSCO), and no mention of the sequencing of the W chromosome. If it was a female sequenced (maybe mention in the results, as well as in methods!) there should be a W chromosome, if there is one. But the results just state “36 scaffolds to 20 chromosomes and the Z chromosome”. If there is a W, then that should appear as a 22nd chromosome with 50% coverage compared to autosomes. Does it have any genes on it? And the Z should also have 50% coverage compared to autosomes.

As it happens, and probably unknown to the authors because of recent pre-publication, a H. charithonia genome has also been sequenced using long-read data, and shows that the W chromosome contains a female-specific UV receptor gene (Chakraborty, M., Lara, A.G., Dang, A., McCulloch, K.J., Rainbow, D., Carter, D., Ngo, L.T., Solares, E., Said, I., Corbett-Detig, R., Gilbert, L.E., Emerson, J.J., & Briscoe, A.D. 2022. Sex-linked gene traffic underlies the acquisition of sexually dimorphic UV color vision in Heliconius butterflies. bioRxiv:2022.2007.2004.498748). So I’d be interested to know: did the authors find the same W chromosome in H. sara, which is reasonably closely related to H. charithonia (assuming the Briscoe lab are prepared to share the genome sequence)? Prior to the Chakraborty et al. paper, no W chromosome had ever been identified in the genus Heliconius. In view of the suggestion by the current authors that the fusions could be Z – A or W – A, it does seem important to mention whether they found a W chromosome at all in the group! Given the assembly of the W in charithonia, are there any signs of a possibly homologous W chromosome in all the other species studied here in addition to the autosomal fusion parts?

I also was disappointed that the manuscript seems poorly written and in places is not clear or grammatical. I am aware that the main authors are not native English speakers and this is therefore understandable. It is also important, I think, that the Colombian authors, in whose country the vast majority of the species occur, should be the ones writing this paper. However, some of the authors are native speakers, and one has recently joined the “Royal Society” and so should be able to speak the King’s English! It seems to me that they ought to be able to help out more than they appear to have done.

Reviewer 1 suggests that in addition to Z-A or W-A fusion, we may actually have both, and that chromosome squashes and counting in both males and females should be done to check. I imagine this is not feasible with Heliconius currently; most chromosome counts have been done only with males. Instead, I would favour a genetics-based approach involving sequencing the brood of offspring of a single female instead, or a new set of long-read assemblies. But these would both be a lot more work, including new field work. I suggest the authors instead say what they have found and what it means, but mine their existing data better in support of their interesting hypotheses.

I’m not a chromosomal expert (although Reviewer 1 is), but I would have thought that the two or three possible hypotheses depend on sex determination in Heliconius, and we do not understand that yet. If the W chromosome contains a gene that determines femaleness, as in Bombyx, then yes, the W-A fusion might create a second Z chromosome from the unfused autosome, so that you have Z1 Z2(the former A) and W-A where the W-A becomes a neo-W. If on the other hand, it is the dosage of the Z chromosome is important in sex-determination, as in Drosophila, a W-A fusion could instead lead to the loss of the W chromosome and the fused part could form a new part of the autosome, once fixed. And of course, we don’t even know whether these putative fusions are fixed.

Here are some critiques I have of the main results.

In Fig. 1 you have found a ML tree for the whole genome, but you make no attempt to accommodate the mixed signal from different loci. I suggest you try to adopt a species tree approach to obtain the species phylogeny, perhaps for males only (to avoid female-specific neo-W chromosomes). As there is no evidence for current gene flow in this group, even an analysis based on the assumption of split without gene flow would be better than this concatenated analysis (which implicitly assumes a lack of recombination at all). A firm hypothesis for the species tree would serve a useful purpose for understanding the comparative analysis of the chromosomal fusions, e.g. in Fig. 6, and for the chromosome-specific contatenated ML trees.

The “phylogenetic”, PCA, Fst and Dxy, whole chromosome heterozygosity, and coverage analyses of males and females in Figs. 2,3,4,5 are all somewhat confusing because the two chromosome morphs in the autosomal fusion chromosomes are presumably lumped together, to give a mixed signal.

In Fig. 3 the “windows clustering by sex” also cluster somewhat by species; and in the “windows clustering by species” there is still some clustering by sex evident. Thus the “grey” and “black” areas in Fig. 3. C are not completely successful in partitioning the sex and species clustering; again you appear to have mixed signal in both grey and black regions of chromosome 4.

In Fig. 4, I assume you use the Hudson & Maddison definition Fst = 1 – (piW/piB), so Fst will be higher if piB (i.e. Dxy) is higher, and lower if piW (i.e. heterozygosity) is higher. So it is a little unclear what Fig. 4 means. The Fst results would be an artefact of the changes in Dxy and/or piW – see Cruickshank & Hahn.

In Figs 5 and 6 I am unconvinced by the heterozygosity and coverage analyses, and want to know more. For example, I’d expect the coverage to be halved in the female when the Z chromosome is used in the females; the “standardized” heterozygosity and mean depth measures don’t seem to show this, though I don’t really know what they do mean (it’s not explained in the figure legend). Also, given that some parts of the fused autosomes in the female appear rather similar in male and female, while most other parts appear divergent between the sexes (high heterozygosity and lowered coverage in females) (Fig. 7), the whole-chromosome heterozygosity and coverage measures in Figs. 5 and 6 seem very poor indicators of what is going on. Also in Fig. 7, in order to accommodate the very high coverage in apparent repeat regions of the female, the scale of mean depth is so small that one can’t distinguish what is going on, with the approx. average 15x coverage, between males and females; in H. sara, the overlap seems complete (as perhaps expected), but only the blue male lines are visible in the figure. Can the scale be increased, with repeat regions annotated but not shown on the scale?

It seems to me that the problem with all these mixed signals is that one gets very little feel for the actual haplotypes underlying these signals. But Illumina data at 15x and higher allows for reasonably accurate heterozygous genotype calling; and given that LD would presumably be very high along the new W version of the autosomal fusion that has no recombination (see reviewer 2’s comment) it seems likely that the differences could be phased for each individual female when you have one or more female and one or more male sequences. See also the comments from reviewer 2.

Thinking about this a little more, it seems to me that providing you can call high confidence heterozygous sites in females, and the SNP is usually homozygous in males (due to the much lower heterozygosity in males as shown in Fig. 7), then one should usually be able to call the female-specific base and the male-specific base in females. For instance, in chromosome 4, if we call these neo Z and neo W haplotypes 4Z and 4W, then for many regions you’d know the sequence for each haplotype. You’d expect the 4W chromosome not to recombine, so it should be fairly easy to pull out the 4W from each female and test for LD decay along the chromosome compared with the 4Z neo-chromosome (can do this with as few as 4 individual diploid sequences, see Heliconius Genome Consortium 2012 supplement). This might be messed up in some SNPs, or in reads with no good SNPs, or in reads with polymorphisms in 4Z as well. But you wouldn’t expect to find any of the divergent 4W haplotypes in males at all, so you should be able to separate a lot of the chromosome’s reads. It might even be possible to use some graph-based assembly program for Illumina (e.g. like DISCOVAR) to at least get slightly longer haplospecific contigs than just the raw reads. However, even the individual SNPs on individual aligned reads should be possible to phase in this way by divergence from the males. Standard methods of LD-based phasing might not work very well, but in principle, I guess that simple-minded divergence based 4W SNPs should work right along the chromosome. You could then construct haploid trees of the 4Z and the 4W chromosomes and separate a large portion of the haploid reads in each individual female. Then all the mixed signals would go away, and the paper would be on firmer ground. You are still unlikely to be able to find the fusion points, but at least you’d have bioinformatic evidence of different 4W haplotypes that show similarity among females, but less similarity between 4W in females and 4Z in males. This would “explain” the heterozygosity results, for example.

On another point, I’d like to see all the precise geo

---

## [Decision Letter · Decision Letter 1]

24 May 2024

Dear Dr Salazar,

We are pleased to inform you that your manuscript entitled "Genomic evidence reveals three sequential W-autosome fusions in Heliconius butterflies" has been editorially accepted for publication in PLOS Genetics. Congratulations!

You will see that there are a number of remaining comments from all three reviewers and from one of us, and we ask that you address those comments with minor revisions during the production process that will not require further editorial evaluation.

Yours sincerely,

James Mallet

Guest Editor

PLOS Genetics

Gregory Barsh

Section Editor

PLOS Genetics

Comments from the guest editor:

I have carefully read the revised manuscript and noted the reviewers’ comments on the revised manuscript. All three reviewers thought the manuscript much improved, and should be published with major, minor, and minor revisions, respectively. I agree and am inclined to accept the paper, but ask the authors to tidy up the manuscript according to all the reviewers’ and indeed my own separate suggestions. I do not want to see the manuscript again before publication, and I think that the editor should accept whatever you turn in. The points I note here refer only to things that I as guest editor thought were worth paying attention to, and also to the suggestions provided by the reviewers. I think paying attention to all of these will enable the authors to spread their ideas more widely, and that these comments will be helpful.

The current manuscript has direct evidence of a W-4-14 chromosomal fusion in H. congener, based on the Hi-C plot of a single female. Amazing what a single HiC plot can do! As the chromosome 4 fusion appears basal to the whole antiochus + sapho + eleuchia clade (e.g. based on haplotype genealogies in Fig. 3), I’m relatively convinced that the chromosomal fusion of 4+W is the ancestral state of this subset of the sara-sapho clade.

Congratulations on some really interesting findings in a well-studied genus of butterflies! I think this is an important paper, and I hope to see it publshed soon!

Reviewer 1 argues for “major revision”, this time, which is better at least than their previous “Reject!”

On my reading, this manuscript is now almost ready to accept for PLoS Genetics, and so I am inclined to suggest that I do not need to see another version. You’re on your own, now! (Apart from the editor-in-chief!). However, the authors should take careful note of all reviewers’ suggestions, and attempt either to answer them or to comply with their recommendations. I’ve also made some additional suggestions of my own below.

Major comments 1: I agree with reviewer 1 that “three sequential fusions” does not seem accurate. In two lineages, an initial fusion of W-4, was followed by a fusion with either chromosome 9 or chromosome 14. Not both, so not sequential in that sense. Please tighten up the language!

Major comments 2: I think I agree with reviewer 1 that the short read data does not provide definitive evidence of the fusions between W and autosomes. A genome assembly was not attempted, so one must rely on the more indirect data from the Hi-C, and the Fst plots, etc. It seems to me that the evidence provided by the authors is strong, but not as convincing as a long read assembly would be. On the other hand, the fact that chromosome 4 is involved throughout does suggest that it is ancestral to the antiochus +sapho + eleuchia group in general.

Major comments 3: “full-length autosomes”. I agree with the reviewer that maybe the term “full-length” is a little inappropriate here. The data provided does not indicate whether the female-specific haplotypes of these autosomes are shorter or longer; but they are likely shorter, given losses will be potentially tolerated given the female will always have an intact homolog (Z2 - 4, Z3 -9/14) of the same chromosome.

Major comments 4: “Is there a reason why the authors chose only the female of H. congener and the male of H. sapho for the Hi-C analyses, but not other species (H. antiochus, H. eleuchia, H. hewitsoni) in the sapho subclade (L354-L355)? The results of the Hi-C analysis suggest the possibility that W-A fusions occur in the female of H. congener and Z-A fusion(s) do not occur in the male of H. sapho. I agree with this assertion. However, the authors claim that W-A fusions but not Z-A fusions have also occurred in other species (H. antiochus, H. eleuchia, H. hewitsoni) of the sapho subclade without specifying how Sex-A fusions have occurred in other species of the sapho subclade. I would therefore suggest rephrasing sentences and/or paragraphs in relation to this claim.”

The reviewer seems to be strictly correct, but I was convinced by the genealogy of haplotypes in e.g. Fig. 3 that the chr 4 – W fusion was ancestral to antiochus, sapho, hewitsoni, eleuchia, and congener. The authors should try to emphasize the evidence for a shared W-4 fusion.

In response to the comment about “...why the authors chose only the female of H. congener and the male of H. sapho for the Hi-C analyses...”, I am guessing it is because all these species are rare and only occur in remote locations.

Minor comments:

“sara/sapho clade” versus “sapho subclade”: I admit that I too was going to comment on this. See below.

Reviewer 2 requests an update on alternative models of Z-A and W-A fusions, which I think is worth repeating. I agree, but given the strong Hi-C evidence of W-A fusion in H. congener, this is perhaps somewhat redundant, now. Please try to sort this out anyway!

I agree with reviewer 2’s concern about the Fst plots.

I also agree that labelling the individual Hi-C plots in Fig. 7 would help the reader! A is sara female, B is sapho male, C is congener female, and D-F are also congener female, D blowup of diploid data for chrs W, 4, and 14: E haploidized W-linked haplotype of same, and F non-W-linked phased haplotype of same, respectively. Labelling within the figures would help the reader!

Reviewer 3:

Please attend to the minor suggestions for phrasing and references and so on!

Guest editor’s suggesions:

The “minor comment” of reviewer 1 on the sara/sapho clade versus the sapho subclade is important, it seems to me! I was myself unclear whenever the sapho subclade was mentioned or the sara sapho clade, which group of species was included. In Fig. 1, it would be sensible to label and name all the subclades of the sara/sapho clade. I suggest “sara-leucadia subclade”, “antiochus subclade”, “sapho-hewitsoni subclade”, and “congener-eleuchia subclade.”

The chromosome numbers are already displayed in Figure 1 (also mention H. charithonia has 21 chromosomes!), but I think additionally, it would be a good idea to display the main hypotheses of W-A fusion here: “antiochus subclade” has W-4 fusion, sapho-hewitsoni subclade has W-4-9 fusion, and congener-eleuchia subclade has W-4-14 fusion.

Line 65. Yes, ref. 31 is now published fully, but also so is ref. 35! Please update references.

Line 69 etc. Please consider these older references on sex chromosomal evolution as recommended by this reviewer.

Lines 82-83: “What are the other two species” I agree with the reviewer that this was a little unclear.

Lines 96-98: The authors should indeed specify the subspecies. The Chiriquí subspecies, H. sara theudela, is morphologically very different from the nominate subspecies H. sara sara from South America. The sequenced specimen is from Panama, and probably belongs to the subspecies H. sara magdalena, although H. sara fulgidus is also possible, as well as theudela in Panama. Which subspecies was sequenced? Be clear! Incidentally, the specimen distribution data by Neil Rosser (https://heliconius-maps.github.io/) are considerably more detailed that the sparse data used for Fig. 1 in this paper.

Line 108: yes, be clear what you think these scaffolds represent.

Fig. S1 legend: Please check this suggestion by Reviewer 1!

Lines 144-146: I don’t fully understand the reviewer’s comment here, but please pay attention to it.

Lines 296-297: I agree with the reviewer that the blue dot in the H. congener plot is most likely chromosome 4 not chromosome 9. Please check this!

My own suggestions include re-working the Fst analyses so that you indicate what you mean by “divergent” – Fst is relative divergence, so often you really seem to think that a lower Fst is due to a higher within species divergence (pi) rather than between species divergence (Dxy).

Also, if you could label the figures within the image portion better, the reader will more quickly appreciate the findings.

Lines 86-7: “the sapho subclade” – a little unclear which species you mean, unless you mean all of the species including antiochus.

Line 87: instead of “the high”, maybe “a larger”?

Line 106: “fewer” and not “less” here.

Line 124: I’d say something like “These putative W scaffolds in H. sara correspond ...”

Lines 134, 137: I think the word “coverage” is used in two different ways, here!

Line 164: I would say “...whole-genome topology was recovered on only eight chromosomes ...”

Fig. 1: I’ve already suggested labelling the subclades for clarity, and also detailing the hypothesized W-A fusion events in each subclade.

Line 197: Should be “homologue” I think, not “homologous”.

Lines 211-213: it’s odd to use lower Fst as evidence for higher divergence within species!

Line 228: “(< 1 female ...homozygous)”: I assume you must mean less than or equal to 1 female? Otherwise why not say all females were heterozygous?

Lines 227-234 and Fig. 3A. I’m a little unclear how this was done, and I don’t know how RELATE works to get the tree, because it’s not possible to draw such a tree with a single SNP. Presumably, the phased haplotype (of what length?) including that SNP was used to draw the tree? Or is this for the whole chromosome? What do the round dots on the branches mean? For each diagnostic W-linked SNP the methods say you polarized the SNP as being derived or ancestral based on comparison with charithonia, but you do not show in Fig. 3 which is derived. Does the brown (female-specific) tree detail the haplotype of the derived SNP?

Line 259: Do you mean Fst between putative sister species here?

Line 265: “lower than average Fst in the sapho subclade”: Here, I think you mean the entire antiochus-sapho-eleuchia subclade with the chr4 fusion? Also, once again, it seems odd to use Fst reduction as a test for an INCREASE in heterozygosity within species due to the W-4 fusion. It’s not obvious that females of different species have less divergent (i.e. Dxy) for W-4 than the rest of the genome, at least from this measurement of Fst, or why this would be expected. If anything one might expect the W-linked autosome to be more divergent, due to relaxed selection that will not remove variants.

p. 24, line 387, Fig. 7. The Hi C figures are key to the argument for W-A linkage, but then only in the H. congener female. One thing I noted (by counting in a blow up of the image) is that the contacts shown clearly delineate 21 chromsomes in H. sara, 56 in the H. sapho male, and 33 in the H. congener female. I was impressed that these are essentially the expected numbers from the previous Keith Brown et al. cytological counts based on testis squashes. I thought you might mention this also! It is very hard to count chromosomes in these squashes; they must be spread out well, and if you have large numbers there is always the possibility that the tiny dot chromosomes overlap on the slide. Therefore, I’ve always somewhat distrusted those earlier karyotype numbers; but no longer!

Line 410: “Increases variation within species and reduces variation between them” – once again, these are Fst results and you’ve not shown reduced % Dxy between species, if I understand correctly, and nor would you necessarily expect it!

Line 425: I’d rephrase: “Our Hi-C data for the H. sapho male (ZZ) showed no evidence for chromosome fusions, ....”

Lines 439-443: These I think are results from the earlier Brown et al. karyotype data, and not from current work, and should be cited as such. You should also highlight that your Hi-C data confirmed some aspects of this work with Hi-C contact maps for three individuals of different species.

Lines 482-483: This phylogenetic conflict with H. hewitsoni is strange, and I wondered if some sort of reticulation at the base of the sapho-antiochus-eleuchia clade might be to blame for this.

Line 485-6: H. sara’s Andean and Amazonian lineages are not dissimilar to what you call the cis- and trans-Andean lineages in H. antiochus, I thought. So H. sara not really “the only species.”

References have been left as blank parentheses in lines 502, 672, 677, 681, and perhaps elsewhere; they should be added.

Lines 593-622: I have already mentioned that I fail to

---

## [Editor Report · Acceptance letter]

24 Jun 2024

PGENETICS-D-23-00346R1 

Genomic evidence reveals three W-autosome fusions in Heliconius butterflies 

Dear Dr Salazar, 

We are pleased to inform you that your manuscript entitled "Genomic evidence reveals three W-autosome fusions in Heliconius butterflies" has been formally accepted for publication in PLOS Genetics! Your manuscript is now with our production department and you will be notified of the publication date in due course.

With kind regards,

Katalin Szabo

PLOS Genetics

On behalf of:
